# VARIATIONAL ADAPTER FOR CROSS-MODAL SIMILARITY REPRESENTATION

## ABSTRACT

The core of vision-language models lies in measuring cross-modal similarity within a unified representation space. However, most image-text matching or multi-class image classification datasets lack fine-grained cross-modal matching annotations, forcing the continuous similarity space into binary classification boundaries. This compression induces false negative samples and significantly impairs the generalization performance of cross-modal tasks. While prior research has attempted to mitigate this by modeling intra-modal ambiguity, it often overlooks inherent annotation flaws, leading to suboptimal uncertainty allocation. To address these challenges, we propose a Variational Adapter for Cross-modal Similarity Representation (VACSR). This approach reformulates image-text matching with fine-grained semantic scarcity as a variational inference problem. It constructs a latent space for cross-modal similarity and uses regularization techniques to mitigate overfitting to binary annotations. Additionally, we introduce a distributional optimization loss to eliminate erroneous gradients caused by false negative samples. We validate the effectiveness of VACSR in image-text retrieval tasks using the COCO Caption dataset and two extended datasets: CxC and ECCV Caption. Furthermore, we conduct comprehensive out-of-distribution evaluations including domain generalization on ImageNet and its variants, as well as base-to-novel generalization across 11 datasets, highlighting VACSR's robust generalization performance in a wide range of real-world situations.

## 1 INTRODUCTION

Unlike traditional visual recognition methods that depend on discrete labels and predefined visual concepts, vision-language models such as Contrastive Language-Image Pre-training (CLIP) Radford et al. (2021); Zhai et al. (2023) directly align images and raw text within a shared representation space, thereby achieving impressive semantic understanding capabilities. These models have been successfully applied to various downstream tasks, including zero-shot classification, cross-modal retrieval, and open-vocabulary detection Pourpanah et al. (2022); Zhou et al. (2022b); Faghri et al. (2017); Chen et al. (2021); Lee et al. (2018); Wei et al. (2025); Wu et al. (2024). However, multimodal paired datasets like MS-COCO Chen et al. (2015), typically use a binary sparse annotation scheme that labels image-text pairs as either "matched" or "mismatched." This labeling approach forces the model to strictly differentiate between all unannotated image-text pairs, potentially overlooking semantic relationships between samples. This issue is particularly evident in fine-tuning scenarios with limited samples, which can significantly compromise the model's generalization performance Chun et al. (2022); Chun (2023); Gao et al. (2024).

The matching relationship between image-text pairs (typically measured by similarity) is inherently complex. As illustrated in Figure 1 (a), consider an image of the Mona Lisa paired with the text "The mysterious smile of the Mona Lisa": from an object co-occurrence perspective, they can be considered matched (the painting contains a smile), but the term "mysterious" represents a subjective perception, making it challenging for models to assign an accurate similarity value along this dimension. Furthermore, this relationship usually varies with similarity in a continuous and gradual pattern. As shown in Figure 1 (b), During the transition from samples annotated as negative to positive, the correspondence between images and texts becomes increasingly evident. For instance, although the third image does not fully match the text "motorcycles", the objects within it are similarly "parked along" the ground in "a bunch". However, the binary sparse annotation lacks

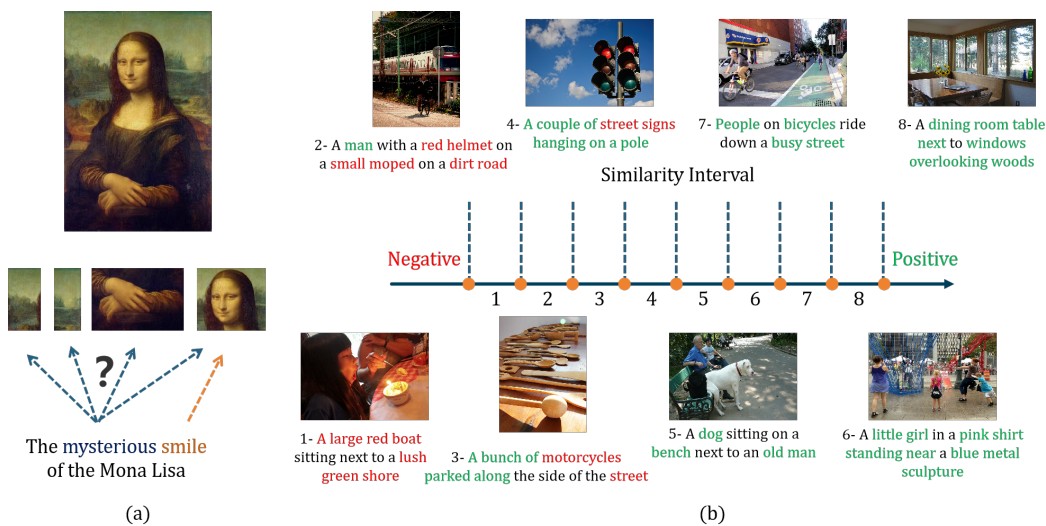

Figure 1: The semantic information embedded in similarity. (a) We take the image of the Mona Lisa and its corresponding textual description as an example to illustrate the matching relationship between image-text data. (b) We computed the similarity of 40,000 sample pairs from the COCO Caption dataset and divided them into eight intervals in ascending order. From each interval, we randomly selected one image-text pair for visual presentation.

the necessary granularity to accurately measure the similarity of image-text pairs. This limitation results in the emergence of false negative sample pairs—i.e., pairs with a certain degree of semantic similarity that are incorrectly regarded as mismatched Li et al. (2023b). Previous studies have shown that false negatives disrupt the semantic consistency of the representation space and limit the model's ability to capture complex matching relationships Chun et al. (2021).

To mitigate the issues caused by binary sparse annotations, existing work has proposed probabilistic embedding methods to model intra-modal ambiguity Chun et al. (2022); Chun (2023); Li et al. (2023a; 2022); Wang et al. (2022); Upadhyay et al. (2023); Ji et al. (2023); Wei et al. (2025). By mapping image and text data into random variables (instead of deterministic vector), these methods expand the potential set of matching results and construct a semantically rich retrieval space, thereby capturing the many-to-many semantic correspondences between image-text data. However, such approaches still rely on binary sparse annotations when computing the cross-modal loss, implying that they address the false negative issue solely from the perspective of the data itself while overlooking annotation errors. (Although PCME++ Chun (2023) introduced a pseudo-labeling strategy, it was simplistic and had limited impact on overall performance.) As a result, the model may assign high uncertainty to any mislabeled false negative sample pairs, even when the semantics are deterministic. Although this strategy can enhance the diversity of retrieval results, it may harm retrieval accuracy when a large number of samples with certain semantics are assigned high uncertainty. Furthermore, intra-modal probabilistic representation requires predicting uncertainty for each modality, leading to a increase in parameter requirements as the number of modalities grows.

To address the aforementioned issues, we reformulate the image-text matching task under sparse annotations as a special form of variational inference problem (VI) and regularize the similarity space to mitigate overfitting to binary annotations. Specifically, we propose a **Variational Adapter for Cross-modal Similarity Representation (VACSR)** to explicitly model the latent space of similarity, treating binary annotations as noisy observations of the underlying true similarity when computing the reconstruction loss. Under this framework, similarity is modeled as a continuous and smooth probability distribution, thereby effectively capturing the fine-grained semantic information missing in binary annotations. Furthermore, to mitigate the impact of binary annotations on false negative samples, we introduce a distributional optimization loss. This loss utilizes the squared distance between the model output and the ground-truth label to optimize the uncertainty in similarity representation. In this process, false negative sample pairs are assigned higher uncertainty to mitigate the impact of erroneous gradients from binary annotations, while informative hard negative sample

pairs are given lower uncertainty to enhance the model's discriminative capability. For more detailed comparisons between VACSR and probabilistic embedding methods, please refer to Appendix H.

Experimental results demonstrate that VACSR offers significant advantages across multiple tasks, including **image-text retrieval, noisy correspondence, domain generalization, and base-to-novel generalization.** Overall, these results show that VACSR effectively enhances model generalization and practical utility in real-world scenarios.

**Contribution**: We propose the Variational Adapter for Cross-modal Similarity Representation (VACSR), a method that addresses the issue of false negative samples by directly modeling the matching relationship between modalities through probabilistic similarity representation. It constructs a semantically rich similarity space and offers favorable interpretability. We evaluate VACSR on three image-text retrieval datasets under both standard and noisy correspondence settings to assess retrieval performance. The method also demonstrates strong out-of-distribution generalization in domain generalization and base-to-novel tasks, showing its broader applicability beyond standard retrieval scenarios.

## 2 PRELIMINARY

For an image-text paired dataset $D = (X, Y)$, the conventional cross-modal alignment strategy employs separate feature encoders $\Phi(\cdot, \theta_{\mathcal{V}})$ and $\Psi(\cdot, \theta_{\mathcal{T}})$ to map paired data $(X_i, Y_j)$ into a d-dimensional space, obtaining their vector representations $\boldsymbol{v}_i = \Phi(X_i, \theta_{\mathcal{V}}), \boldsymbol{t}_j = \Psi(Y_j, \theta_{\mathcal{T}})$, where $\boldsymbol{v}_i, \boldsymbol{t}_j \in \mathbb{R}^d$. A pairwise loss is then typically applied to maximize the similarity of positive pairs while minimizing that of negative pairs, and the cross-modal similarity is measured using cosine similarity. This process is regarded as supervised learning and employs binary annotations to optimize the loss. (Hereafter, we make no distinction between the terms "sample" and "sample pair".)

As mentioned earlier, the binary annotation forcibly separates the continuous similarity space, which can lead to the emergence of false negative samples. Therefore, a well-designed pairwise loss should exhibit tolerance towards semantically similar samples. However, common contrastive loss and sigmoid losse Zhai et al. (2023) fail to handle this issue effectively. We first compute the gradient of the pairwise loss $\mathcal{L}(\mathbf{S})$ with respect to the output $B \times B$ similarity matrix $\boldsymbol{S}$ at the $t$-th iteration:

$$\frac{\partial \mathcal{L}(\boldsymbol{S})}{\partial S_{i,j}} = \sum_{i=1}^{B}[(\sum_{j \neq i, (i,j) \in c}^{B} \frac{\partial \mathcal{L}(\boldsymbol{S})}{\partial S_{i,j}} + \sum_{j=i}^{B} \frac{\partial \mathcal{L}(\boldsymbol{S})}{\partial S_{i,j}}) + \sum_{j \neq i, (i,j) \notin c}^{B} \frac{\partial \mathcal{L}(\boldsymbol{S})}{\partial S_{i,j}}] \tag{1}$$

where $S_{i,j} \in c$ denotes that the image text pair $(X_i, Y_j)$ belongs to the set of false negative samples. In practice, false negative samples exhibit potential semantic relations to positive samples yet provide gradients in the opposite direction. Therefore, for an image $X_i$, we define the gradient difference between positive and false negative samples as $r_i = |\sum_{j=i}^{B} \frac{\partial \mathcal{L}(\boldsymbol{S})}{\partial S_{i,j}}| - |\sum_{j \neq i, (i,j) \in c}^{B} \frac{\partial \mathcal{L}(\boldsymbol{S})}{\partial S_{i,j}}|$, which reflects the tolerance of the loss function to false negative samples. A smaller $r_i$ indicates greater difficulty for the model in learning semantically consistent representations. Meanwhile, the gradient of positive samples is also suppressed. When $r_i < 0$ the positive samples can no longer constrain the optimization direction of false negative samples, and semantically similar samples are forcibly separated, thereby disrupting the underlying semantic structure.

When the pair-based loss adopts contrastive loss $\mathcal{L}_{contrast} = -\log \frac{exp(S_{i,i}/\tau)}{\sum_{j \neq i}^{m} exp(S_{i,j}/\tau) + exp(S_{i,i}/\tau)}$, where $\tau$ represents the temperature coefficient, the corresponding $r_i$ can be computed as follows:

$$r_i = \frac{1}{\tau} \frac{\sum_{j \neq i, (i,j) \notin c} exp(S_{i,j}/\tau)}{\sum_{j \neq i}^{n} exp(S_{i,j}/\tau) + exp(S_{i,i}/\tau)} \tag{2}$$

From Equation 2, we draw the following conclusions: (1) Since $r_i$ is always greater than 0, $\mathcal{L}_{contrast}$ does not completely disrupt the semantic consistency of representations; (2) The magnitude of $r_i$ equals the sum of all softmax-normalized negative samples multiplied by the reciprocal of $\tau$. Here, $\tau$ controls the smoothness of the distribution, a smaller $\tau$ causes an rapid reduction in the contribution of negative samples Wang & Liu (2021), thereby increasing the disruption to semantic consistency; a larger $\tau$ increases $r_i$ but can hinder the learning of separable features Wang & Liu (2021). Thus, $\mathcal{L}_{contrast}$ must balance $r_i$ and $\tau$.

When the pair-based loss adopts sigmoid loss: $\mathcal{L}_{sigmoid} = -\log \frac{1}{1+exp(z_{i,j}(-aS_{i,j}+b))}$, where $z_{i,j} = 1$ for positive samples and $z_{i,j} = -1$ otherwise. The corresponding $r_i$ can be computed as follows:

$$r_i = |a|[(1 - P_{i,i}) - \sum_{S_{i,j} \in c} P_{i,j}] \tag{3}$$

where $P_{i,j} = sigmoid(-aS_{i,j} + b)$. We make the following observations: (1) In this case, $r_i$ does not require softmax normalization. When the similarity of positive samples is fixed, the tolerance of $\mathcal{L}_{sigmoid}$ depends solely on the false negative samples themselves. (2) Since $r_i$ can be less than 0, $\mathcal{L}_{sigmoid}$ carries the risk of losing semantic information. Thus, it is necessary to employ suitable scalar parameters $a$ and $b$, which aligns with the empirical observations in SigLIP. In Appendix D, we compare the sensitivity of the sigmoid loss and contrastive loss to scaling parameters and temperature coefficients, further elucidating the impact of binary annotations on retrieval performance.

In summary, within the binary annotation framework, contrastive loss is sensitive to softmax normalization and requires careful tuning of the temperature coefficient. Although the tolerance of sigmoid loss is relatively independent, it necessitates the introduction of scalar parameters to mitigate the risk of $r_i < 0$. Based on this analysis, we propose modeling the similarity $S_{i,j}$ in a latent space using a variational adapter and introduce a distributional optimization loss. This loss incorporates the uncertainty of the latent space into the reconstruction loss computation via the reparameterization trick, thereby eliminating the adverse gradient effects of false negative samples. Furthermore, the reconstruction loss also maintains the independence of $r_i$ and models a continuous similarity space, thereby preserving the integrity of the semantic structure.

## 3 METHODOLOGY

The overall architecture of VACSR is illustrated in Figure 2. We will detail the cross-modal similarity representation structure in Section 3.1, and elaborate on how to optimize the latent variable distribution for better modeling of false negative samples in Section 3.2.

### 3.1 SIMILARITY REPRESENTATION

We propose a variational adapter to model the representation of cross-modal similarity, uncovering rich matching relationships that be overlooked by binary annotations. As shown in Figure 2, a batch of image-text pairs are first encoded by a CLIP encoder to obtain their respective feature representations $\boldsymbol{V} = [\boldsymbol{v}_1, \boldsymbol{v}_2, ...\boldsymbol{v}_B] \in \mathbb{R}^{B \times d}$ and $\boldsymbol{T} = [\boldsymbol{t}_1, \boldsymbol{t}_2, ...\boldsymbol{t}_B] \in \mathbb{R}^{B \times d}$. Feature interaction is then achieved through the Hadamard product $\boldsymbol{S} = \boldsymbol{V} \odot \boldsymbol{T} \in \mathbb{R}^{B \times B \times d}$, where $\boldsymbol{s}_{i,j} \in \boldsymbol{S}$ can be regarded as a vector representation of similarity. Unlike the direct use of cosine similarity, the Hadamard product performs element-wise multiplication, which preserves the dimensionality of the interactive feature, thereby supporting subsequent encoding procedures. Subsequently, the latent representation $\mathbf{z}_{i,j}$ of $\boldsymbol{s}_{i,j}$ is constructed via the variational adapter. The design of this variational adapter references the variational autoencoder (VAE) Kingma & Welling (2013), and we optimize the model by maximizing the Evidence Lower Bound (ELBO).

$$ELBO = \mathbb{E}_{p_\phi(\mathbf{z}_{i,j}|\boldsymbol{s}_{i,j})}[\log q_\theta(\hat{S}_{i,j}|\mathbf{z}_{i,j})] - \text{KL}[p_\phi(\mathbf{z}_{i,j}|\boldsymbol{s}_{i,j})||q(\mathbf{z}_{i,j})] \tag{4}$$

where $\hat{S}_{i,j}$ denotes the true similarity containing fine-grained information, which we temporarily replace with the binary label $\hat{y}_{i,j}$. We will introduce how to correct $\hat{y}_{i,j}$ in Section 3.2. Prior $q(\mathbf{z}_{i,j})$ is set to the standard normal distribution. $p_\phi$ and $q_\theta$ denote the parameterized similarity representation encoder and decoder composed of multilayer perceptrons (MLPs), respectively. To prevents the unimodal nature of a Gaussian distribution from restricting the model's capacity to learn complex semantic representations Bai et al. (2022); Ye et al. (2025), we approximate the $p_\phi(\mathbf{z}_{i,j}|\boldsymbol{s}_{i,j})$ as a two-component Gaussian mixture model. Thus, the regularization term $\mathcal{L}_{KL}$ can be computed as:

$$\alpha_1 \text{KL}[\mathcal{N}(\mu_1(\boldsymbol{s}_{i,j}), \sigma_1(\boldsymbol{s}_{i,j}))||\mathcal{N}(0,1)] + \alpha_2 \text{KL}[\mathcal{N}(\mu_2(\boldsymbol{s}_{i,j}), \sigma_2(\boldsymbol{s}_{i,j}))||\mathcal{N}(0,1)] \tag{5}$$

here, $\mu_1, \mu_2, \sigma_1, \sigma_2 \in \mathbb{R}^d$ are derived from the $\mu$ and $\log \sigma^2$ heads corresponding to two different Gaussian components in $p_\phi$, and $\alpha$ is learnable mixing coefficient with the constraint $\alpha_1 + \alpha_2 = 1$.

Figure 2: Overview of our proposed model: Image and text features first interact through the Hadamard product to generate similarity vector representations, which are then input into a variational adapter composed of an encoder and a decoder. The encoder predicts the mean ($\mu$) and log-variance ($log\sigma^2$) for each similarity vector, mapping the input to a Gaussian mixture latent distribution, where each Gaussian component is regularized to a standard normal distribution. Using the reparameterization trick, latent variables are sampled and subsequently reconstructed into a similarity matrix by the decoder. The model is optimized by minimizing the reconstruction loss ($\mathcal{L}_{recon}$) between the predicted similarities and the binary labels, while a distributional optimization loss ($\mathcal{L}_\sigma$) is introduced to adaptively calibrate the uncertainty of the latent representations.

The VAE approximates the generative model $q_\theta(\hat{S}_{i,j}|\mathbf{z}_{i,j})$ as a Gaussian distribution, and the reconstruction term $\mathcal{L}_{recon}$ is formulated as the negative log-likelihood under this Gaussian assumption:

$$\mathbb{E}_{p_\phi(\mathbf{z}_{i,j}|\boldsymbol{s}_{i,j})}[\log q_\theta(\hat{S}_{i,j}|\mathbf{z}_{i,j})] = -\mathbb{E}_{p_\phi(\mathbf{z}_{i,j}|\boldsymbol{s}_{i,j})}[\log(\frac{1}{\sqrt{2\pi\sigma^2(\mathbf{z}_{i,j})}}\exp(-\frac{(\hat{y}_{i,j}-\mu(\mathbf{z}_{i,j}))^2}{2\sigma^2(\mathbf{z}_{i,j})}))]$$

$$= \mathbb{E}_{p_\phi(\mathbf{z}_{i,j}|\boldsymbol{s}_{i,j})}[\frac{1}{2\sigma^2(\mathbf{z}_{i,j})}||\hat{y}_{i,j}-\mu(\mathbf{z}_{i,j})||^2 + \log\sigma(\mathbf{z}_{i,j}) + \frac{1}{2}\log 2\pi]$$

$$(6)$$

Ignoring the constant term, this loss is equivalent to the mean squared error (MSE) loss. Here, $\mu(\mathbf{z}_{i,j}) \in \mathbb{R}^1$ is obtained from the decoder $q_\theta$, normalized by a sigmoid function, and regarded as the final similarity score. The variance $\sigma^2(\mathbf{z}_{i,j})$ is typically fixed to 1. For the latent variable $\mathbf{z}_{i,j}$, we first use the Gumbel-Softmax trick to reparameterize the binary Gaussian component selection process, obtaining a differentiable approximate one-hot vector. We then apply the standard Gaussian reparameterization trick to sample from each Gaussian component. The final sampling point is obtained through a one-hot vector weighted sum of the sampling results using the one-hot vector.

## 3.2 OPTIMIZATION OF UNCERTAINTY IN LATENT VARIABLES

As mentioned in Section 1, the binary annotation $\hat{y}$ lacks sufficient fine-grained semantic information. Consequently, directly substituting $\hat{S}_{i,j}$ with $\hat{y}$ can result in incorrect gradients for false negative samples. We address this issue by managing the uncertainty of the latent variable. Given that $\mathbf{z}_{i,j}$ is sampled from $p_\phi(\mathbf{z}_{i,j}|\boldsymbol{s}_{i,j})$, we expand Equation 6 as follows:

$$\frac{1}{2}\mathbb{E}_{p_\phi(\mathbf{z}_{i,j}|\boldsymbol{s}_{i,j})}[||\hat{y}-\mu(\mathbf{z}_{i,j})||^2] = \frac{1}{2}||\hat{y}-\mu[\hat{\mu}(\boldsymbol{s}_{i,j})+\varepsilon\cdot\hat{\sigma}(\boldsymbol{s}_{i,j})]||^2 \qquad (7)$$

here $\hat{\mu}(\boldsymbol{s}_{i,j}), \hat{\sigma}(\boldsymbol{s}_{i,j})$ are derived from the currently sampled Gaussian components, and sampling is performed only once. $\varepsilon$ is Gaussian random noise. Let us consider two extreme cases: when $\hat{\sigma}(\boldsymbol{s}_{i,j}) \to 0$, $\mathcal{L}_{recon}$ optimizes $\hat{\mu}(\boldsymbol{s}_{i,j})$ towards binary results. Conversely, when $\hat{\sigma}(\boldsymbol{s}_{i,j}) \to \infty$, $\hat{\mu}(\boldsymbol{s}_{i,j})$ becomes negligible, and $\mu[\hat{\mu}(\boldsymbol{s}_{i,j})+\varepsilon\cdot\hat{\sigma}(\boldsymbol{s}_{i,j})]$ turns into Gaussian random noise. In this

case, $\mathcal{L}_{recon}$ no longer influences $\hat{\mu}(\boldsymbol{s}_{i,j})$. Thus, we can control the optimization strength of the binary annotations by adjusting $\hat{\sigma}(\boldsymbol{s}_{i,j})$ and only decode $\hat{\mu}(\boldsymbol{s}_{i,j})$ as the final similarity output.

Based on this analysis, in the $t$ iteration, we should assign lower uncertainty to positive samples and informative negative samples (typically hard negative samples Wang & Liu (2021)) to enhance prediction accuracy, while assigning higher uncertainty to false negative samples to mitigate the interference caused by erroneous annotations. Accordingly, we introduce an additional distributional optimization loss for the variance $\hat{\sigma}(\boldsymbol{s}_{i,j})$:

$$\mathcal{L}_{\sigma} = \frac{1}{2\hat{\sigma}(\boldsymbol{s}_{i,j})}||\hat{y} - \mu(\mathbf{z}_{i,j})||^2 + \log\hat{\sigma}(\boldsymbol{s}_{i,j}) \tag{8}$$

We truncate the gradient of the term $||\hat{y} - \mu(\mathbf{z}_{i,j})||^2$ and use it solely as a weighting coefficient to optimize the variance term. By differentiating $\mathcal{L}_{\sigma}$ and setting the derivative to zero, we obtain:

$$\frac{\partial\mathcal{L}_{\sigma}}{\partial\hat{\sigma}(\boldsymbol{s}_{i,j})} = -\frac{1}{\hat{\sigma}^3(\boldsymbol{s}_{i,j})}||\hat{y} - \mu(\mathbf{z}_{i,j})||^2 + \frac{1}{\hat{\sigma}(\boldsymbol{s}_{i,j})} = 0 \tag{9}$$

$$\hat{\sigma}^2(\boldsymbol{s}_{i,j}) = ||\hat{y} - \mu(\mathbf{z}_{i,j})||^2$$

This indicates that the optimized variance equals the squared distance between the model's output similarity and the true label. Samples closer to the label exhibit smaller variance, while those farther away have larger variance, thus achieving our goal of assigning different levels of uncertainty to different samples. Furthermore, since the loss only constrains the variance within the $[0, 1]$ interval, we additionally apply a sigmoid function to normalize the output $\log\sigma^2$ from $p_{\phi}$. This approach extends the domain of variance to $[0, +\infty]$ while preserving its range. Then, we compute the loss for positive samples and hard negative samples separately:

$$\mathcal{L}_{\sigma}^P = \frac{1}{2\hat{\sigma}^2(\mathbf{s}_{i,i})}||1 - \mu(\mathbf{z}_{i,i})||^2 + \log\hat{\sigma}(\mathbf{s}_{i,i})$$

$$\mathcal{L}_{\sigma}^N = \max_j[\frac{1}{2\hat{\sigma}^2(\mathbf{s}_{i,j})}||1 - \mu(\mathbf{z}_{i,j})||^2 + \log\hat{\sigma}(\mathbf{s}_{i,j})] \tag{10}$$

Here, we still set $\hat{y} = 1$ for hard negative samples. In fact, cross-modal alignment relies on hard negative samples to better distinguish between positive and negative pairs Faghri et al. (2017). If $\hat{y} = 0$ were used, hard negative samples with similarity close to 1 would be assigned high uncertainty, which is detrimental to model's optimization.

**Objective function**: The final objective function is defined by optimizing a weighted sum of multiple losses, with hyperparameters $\alpha$, $\beta$ and $\gamma$ serving as weighting coefficients. In all experiments, the hyperparameters are uniformly set to $\alpha = 0.0005, \beta = \gamma = 1$:

$$\mathcal{L} = \alpha\mathcal{L}_{KL} + \beta[\mathcal{L}_{recon} + \gamma(\mathcal{L}_{\sigma}^P + \mathcal{L}_{\sigma}^N)] \tag{11}$$

Meanwhile, we adopt a straight-through estimator (STE) Jacob et al. (2018) to address the gradient anomaly arising from the application of the MSE loss in classification tasks. Details can be found in Appendix A.1.

# 4 EXPERIMENT

## 4.1 EXPERIMENTAL SETUP

We assess the generalization performance of the VACSR model on two downstream tasks: image-text retrieval and out-of-distribution generalization. Specifically, for the image-text retrieval task, we additionally introduce a **noisy correspondence** setting. For out-of-distribution scenarios, we consider two types: **base-to-novel generalization** and **domain generalization.** Detailed experimental settings are provided in Appendix B.1 and B.2.

## 4.2 MAIN RESULTS

**Image-Text Retrieval**: Tables 1 and 2 present the performance comparison between VACSR and other methods on the image-text retrieval task. We select representative baselines including P2RM

Table 1: The performance comparison between our model and other approaches on the COCO dataset. We present the Recall@K and RSUM metric results, including both the 1K test setting (averaged over 5-fold test datasets) and the 5K test setting. The best results are highlighted in bold.

| Method | 1K Test Images | | | | | | 5K Test Images | | | | | |
| | Image-to-Text | | | Text-to-Image | | | Image-to-Text | | | Text-to-Image | | |
| | R@1 | R@5 | R@10 | R@1 | R@5 | R@10 | R@1 | R@5 | R@10 | R@1 | R@5 | R@10 |
|---|---|---|---|---|---|---|---|---|---|---|---|---|
| *CLIP ViT-B/32* | | | | | | | | | | | | |
| P2RM | 78.6 | 96.1 | 98.6 | 67.5 | 92.3 | 96.8 | 56.6 | 83.5 | 90.9 | 46.3 | 74.9 | 84.4 |
| DAA | 79.8 | 96.5 | 98.9 | 67.4 | 91.9 | 96.5 | 59.8 | 85.0 | 92.0 | 46.1 | 74.7 | 84.2 |
| PCME | 80.1 | 96.6 | 98.7 | 67.6 | 92.1 | 96.9 | 59.9 | 85.8 | 92.3 | 46.1 | 75.0 | 84.6 |
| PCME++ | 81.6 | 97.0 | 99.0 | 69.2 | 92.8 | 97.1 | 62.1 | 87.0 | 93.0 | 48.1 | 76.5 | 85.4 |
| **VACSR** | **84.2** | **97.2** | **99.0** | **70.3** | **93.3** | **97.4** | **66.5** | **88.3** | **93.9** | **49.8** | **77.7** | **86.3** |
| *CLIP ViT-B/16* | | | | | | | | | | | | |
| P2RM | 78.3 | 96.2 | 98.7 | 69.2 | 93.0 | 97.2 | 56.8 | 84.3 | 91.5 | 48.1 | 76.6 | 85.7 |
| DAA | 46.1 | 76.8 | 87.6 | 41.3 | 73.4 | 85.1 | 24.3 | 49.9 | 62.7 | 22.4 | 47.1 | 59.1 |
| PCME | 83.6 | 97.7 | 99.3 | 72.0 | 93.9 | 97.7 | 65.3 | 89.2 | 94.5 | 51.2 | 79.1 | 87.5 |
| PCME++ | 85.3 | 97.9 | 99.3 | 73.4 | 94.4 | 97.8 | 68.7 | 90.1 | 95.0 | 53.4 | 80.3 | 88.3 |
| **VACSR** | **87.4** | **98.2** | **99.4** | **74.3** | **94.6** | **97.9** | **72.2** | **91.1** | **95.4** | **54.5** | **81.1** | **88.7** |

Table 2: The performance comparison between our model and other approaches on the EC and CXC datasets. We present the Recall@K, R-P and mAP@R metric results, the best results are highlighted in bold.

| Method | ECCV Caption | | | | | | CxC | | | | | |
| | Image-to-Text | | | Text-to-Image | | | Image-to-Text | | | Text-to-Image | | |
| | R@1 | R-P | mAP@R | R@1 | R-P | mAP@R | R@1 | R@5 | R@10 | R@1 | R@5 | R@10 |
|---|---|---|---|---|---|---|---|---|---|---|---|---|
| *CLIP ViT-B/32* | | | | | | | | | | | | |
| P2RM | 72.2 | 41.7 | 30.2 | 89.5 | 55.5 | 47.6 | 58.1 | 85.5 | 92.2 | 48.4 | 77.2 | 86.2 |
| DAA | 75.9 | 42.3 | 31.2 | 88.1 | 55.7 | 47.3 | 61.5 | 86.8 | 93.3 | 48.2 | 76.9 | 86.1 |
| PCME | 74.9 | 42.3 | 31.2 | 88.0 | 55.5 | 47.1 | 61.5 | 87.5 | 93.5 | 48.0 | 77.3 | 86.4 |
| PCME++ | 76.6 | 43.4 | 32.3 | 89.5 | 55.9 | 47.8 | 63.5 | 88.4 | 94.0 | 50.1 | 78.5 | 87.1 |
| **VACSR** | **80.6** | **43.8** | **33.2** | **90.4** | **56.3** | **48.1** | **67.8** | **89.6** | **94.9** | **51.6** | **79.7** | **88.0** |
| *CLIP ViT-B/16* | | | | | | | | | | | | |
| P2RM | 72.9 | 42.2 | 30.6 | 88.5 | 56.8 | 48.8 | 58.5 | 86.0 | 92.7 | 50.0 | 78.7 | 87.3 |
| DAA | 40.3 | 22.4 | 12.4 | 60.0 | 38.9 | 29.0 | 26.4 | 53.9 | 67.1 | 24.3 | 50.3 | 62.5 |
| PCME | 79.1 | 44.0 | 33.2 | 89.5 | 56.5 | 48.7 | 66.8 | 90.5 | 95.4 | 53.1 | 80.9 | 88.9 |
| PCME++ | 81.6 | 45.1 | 34.5 | 91.4 | 57.2 | 49.7 | 69.9 | 91.3 | 95.7 | 55.2 | 82.0 | 89.7 |
| **VACSR** | **84.9** | **45.5** | **35.4** | **92.2** | **57.5** | **49.7** | **73.3** | **92.0** | **96.2** | **56.3** | **82.8** | **90.1** |

(MM) Wang et al. (2022), DAA (NeurIPS) Li et al. (2022), PCME (CVPR) Chun et al. (2021), and PCME++ (ICLR) Chun (2023), all of which mitigate the false negative sample issue by modeling intra-modal ambiguity. Notably, our method employs a simple two-layer MLP structure instead of the Transformer-based uncertainty prediction used in PCME++, reducing the total parameter count to only 48.3% of PCME++. Experimental results demonstrate that VACSR achieves the best performance across all evaluation metrics under different backbone networks. Specifically, compared to PCME++, VACSR exhibits superior performance on the mAP@R and R@1 metrics of the EC dataset, validating its capability to understand diverse semantic information. Furthermore, VACSR achieves significant improvements in R@1 across all evaluated datasets. Taking ViT-B/32 as an example, it achieves improvements of 3.2%, 7.1%, 5.2%, and 6.8% on the respective datasets in the image-to-text retrieval direction. This indicates that VACSR can assign appropriate uncertainty to different types of samples, thereby ensuring retrieval accuracy. Moreover, when scaling the backbone network from ViT-B/32 to ViT-B/16, VACSR maintains performance improvements without any adjustments to the variational adapter. This demonstrates that the variational adapter structure can effectively model the semantic distribution of false negative samples, unaffected by the complexity of the backbone network.

Table 3: Noisy correspondence results using the ViT-B/32 backbone are shown. All reported metrics represent the average performance over both image-to-text and text-to-image retrieval directions.

| Noise ratio | Method | ECCV Caption | | | CxC | COCO | | |
|---|---|---|---|---|---|---|---|---|
| | | mAP@R | R-P | R@1 | R@1 | 1K R@1 | 5K R@1 | RSUM |
| 20% | VSE∞ | 37.0 | 46.3 | 79.7 | 53.6 | 72.0 | 51.8 | 518.6 |
| | DAA | 6.7 | 12.5 | 18.5 | 7.0 | 15.3 | 6.0 | 212.8 |
| | PCME | 37.6 | 47.6 | 79.2 | 50.6 | 70.3 | 48.7 | 520.7 |
| | NCR | 35.9 | 46.0 | 78.0 | 50.6 | 70.1 | 48.8 | 518.6 |
| | BiCro | - | - | - | - | 71.3 | - | 523.2 |
| | PCME++ | 37.7 | 47.6 | 80.0 | 52.2 | 71.6 | 50.4 | 524.6 |
| | NPC | - | - | - | - | 73.1 | 53.8 | 529.8 |
| | VACSR | **40.1** | **49.6** | **83.9** | **58.7** | **76.4** | **57.1** | **539.0** |
| 50% | VSE∞ | 18.0 | 28.5 | 43.7 | 20.7 | 39.2 | 19.1 | 394.1 |
| | DAA | 0.3 | 0.8 | 1.0 | 0.3 | 0.8 | 0.2 | 20.9 |
| | PCME | 35.2 | 45.5 | 75.7 | 46.3 | 66.6 | 44.4 | 508.0 |
| | NCR | 34.0 | 44.3 | 75.1 | 47.3 | 66.8 | 45.5 | 508.5 |
| | PCME++ | 35.7 | 45.8 | 76.3 | 47.4 | 67.6 | 45.5 | 511.0 |
| | NPC | - | - | - | - | 71.3 | 51.9 | 523.4 |
| | VACSR | **39.5** | **49.1** | **82.8** | **57.2** | **75.1** | **55.6** | **534.2** |

Table 4: Performance comparison of various methods on base-to-novel generalization across 11 datasets, using CLIP ViT-B/16 as the encoder backbone. Results are averaged over all 11 datasets.

| | BASE | NEW | H |
|---|---|---|---|
| ZERO-SHOT | 69.34 | 74.22 | 71.70 |
| CoCoOp | 80.47 | 71.69 | 75.83 |
| CLIPOOD | 83.9 | 74.5 | 78.9 |
| MaPLe | 82.28 | 75.14 | 78.55 |
| CoPrompt | 84.00 | **77.23** | 80.48 |
| MMA | 83.20 | 76.80 | 79.87 |
| MMRL | 85.68 | 77.16 | **81.20** |
| VACSR | **85.74** | 76.08 | 80.37 |

**Noisy correspondence**: As shown in Table 3, by injecting varying proportions of Noisy Correspondence into the training data, we systematically assess the model's adaptability and stability under noisy conditions Huang et al. (2021). Additionally, we include three representative Noisy Correspondence learning methods—NCR Huang et al. (2021), BiCro Yang et al. (2023), and NPC Zhang et al. (2024). Experimental results demonstrate that the proposed VACSR method exhibits superior noise robustness across various noise levels. Under a 20% noise ratio, VACSR outperforms the next-best method PCME++ by 6.3%, 4.2%, and 4.9% in mAP@R, R-P, and R@1 of the ECCV Caption dataset, respectively; achieves a 12.5% improvement in R@1 on the CxC dataset; and delivers 4.5%, 6.1%, and 1.7% gains over NPC in 1K R@1, 5K R@1, and RSUM on the COCO dataset. At a high noise ratio of 50%, VACSR maintains leading performance, while other methods drop sharply. It achieves improvements of 10.6%, 7.2%, 8.5%, 20.7%, 5.3%, 7.1%, and 2.1% over the next-best methods across datasets. In summary, VACSR effectively mitigates the impact of noisy annotations and demonstrates strong robustness for real-world applications.

**Base-to-Novel Generalization**: The lack of semantic information in binary sparse annotations may also limit the model's generalization ability to unseen categories. Therefore, we further conduct base-to-novel generalization experiments to evaluate whether similarity-based representations can learn more generalizable semantic features. Although the VACSR method is not specifically designed for out-of-distribution generalization tasks, its average performance across 11 datasets demonstrates competitive generalization capability. As shown in Table 4, VACSR achieves a harmonic mean (H) score of 80.37, outperforming most baseline methods (e.g., CoCoOp at 75.83 and MaPLe at 78.55), indicating strong overall performance. Notably, VACSR attains the best performance on base class recognition with a score of 85.74, demonstrating its advantage in retaining discriminative ability on seen categories. Although the accuracy on novel classes is slightly lower than some methods specialized in generalization (e.g., CoPrompt and MMRL), VACSR maintains a

Table 5: Performance comparison of different methods on ImageNet and its variants. We employ Clip ViT-B/16 as the encoder backbone. The best results are highlighted in bold.

| METHOD | IN-DISTRIBUTION | OUT-OF-DISTRIBUTION | | | | |
|---|---|---|---|---|---|---|
| | IMAGENET | V2 | S | A | R | AVG. |
| ZERO-SHOT | 66.7 | 60.8 | 46.1 | 47.8 | 74.0 | 57.2 |
| COCOOP | 71.0 | 64.1 | 48.8 | 50.6 | 76.2 | 59.9 |
| CLIPOOD | 71.6 | 64.9 | 49.3 | 50.4 | 77.2 | 60.4 |
| MaPLe | 70.7 | 64.1 | 49.2 | 50.9 | 77.0 | 60.3 |
| CoPrompt | 70.8 | 64.3 | 49.4 | 50.5 | 77.5 | 60.4 |
| MMA | 71.0 | 64.3 | 49.1 | 51.1 | 77.3 | 60.5 |
| MMRL | 72.0 | 64.5 | 49.2 | 51.2 | 77.5 | 60.6 |
| **VACSR** | **74.3** | **65.7** | **49.7** | **52.4** | **78.4** | **61.6** |

Table 6: Ablation study on the EC, CXC and COCO datasets. All reported metrics represent the average performance over both image-to-text and text-to-image retrieval directions.

| Configuration | ECCV Caption | | | CxC | COCO | | |
|---|---|---|---|---|---|---|---|
| | mAP@R | R-P | R@1 | R@1 | 1K R@1 | 5K R@1 | RSUM |
| w/ All Components | **40.7** | **50.1** | **85.5** | 59.7 | 77.2 | 58.1 | 541.4 |
| w/o $\mathcal{L}_{KL}$ | 40.1 | 49.4 | 85.3 | 60.4 | 77.3 | 58.8 | **541.5** |
| w/o *sigmoid* | 40.1 | 49.4 | 85.5 | 60.4 | 77.2 | 58.8 | 541.1 |
| w/o $\mathcal{L}_{\sigma}$ | 39.8 | 49.1 | 84.9 | **60.7** | **77.3** | **59.1** | 541.1 |
| w/o GMM | 39.6 | 49.0 | 84.1 | 59.8 | 77.0 | 58.1 | 540.7 |
| w/o All Components (Baseline) | 39.3 | 48.7 | 83.1 | 57.3 | 75.5 | 55.6 | 537.0 |

well-balanced performance overall, without significant drops in novel class accuracy due to overfitting on base classes. By modeling cross-modal semantic distributions through variational inference, VACSR can partially overcome reliance on specific category labels, thus enabling generalization to semantically related unseen classes. Although there remains room for improvement compared to methods specifically designed for out-of-distribution generalization, these findings offer valuable insight into the relationship between similarity modeling and generalization ability. Please refer to Appendix D for detailed results on each dataset.

**Domain Generalization**: CLIP extends its semantic understanding capability to classification tasks by computing the similarity between handcrafted text prompts (e.g., "a photo of a <class>") and different visual categories . However, classification datasets like ImageNet typically employ fixed annotations for visual categories, overlooking the potential fine-grained semantic relationships between different visual concepts and text. Therefore, we further conducted domain generalization experiments to evaluate the cross-task generalization capability of VACSR and the application potential of its similarity representation. We compared our method with prompt-based learning approaches (e.g., COCOOP (CVPR) Zhou et al. (2022a), MaPLe (CVPR) Khattak et al. (2023), CoPrompt (ICLR) Roy & Etemad (2023)), adapter-based and fine-tuning methods (e.g., MMA (CVPR) Yang et al. (2024), CLIPOOD (ICML) Shu et al. (2023)), and methods that also model shared representation spaces (e.g., MMRL (CVPR) Guo & Gu (2025)). As shown in Table 5, VACSR achieved the best performance across all datasets, demonstrating its generalization capability and robustness to domain shifts. This result underscores that classification tasks also necessitate modeling a more continuous latent representation space to capture the inherent semantic of visual concepts effectively.

## 4.3 ABLATION STUDY

Table 6 demonstrates the effectiveness of $\mathcal{L}_{KL}$, $\mathcal{L}_{\sigma}$, the sigmoid function used in $\mathcal{L}_{\sigma}$, and the adoption of a Gaussian Mixture Model (GMM) prior. The baseline uses a Generalized Pooling Operator (GPO) and sigmoid loss to fine-tune CLIP directly. Experimental results show that the GMM prior has the most significant impact on overall performance, particularly in the EC dataset. This suggests that the unimodal nature of a single Gaussian prior is inadequate for fully modeling the latent space of similarity, further supporting the presence of complex semantic information in similarity representations. Compared to $\mathcal{L}_{KL}$, $\mathcal{L}_{\sigma}$ has a more pronounced effect on the EC datasets, indicating that

$\mathcal{L}_{\sigma}$ effectively reduces the influence of binary annotations. Removing the sigmoid function leads to a decrease of 1.5% in mAP@R and 1.4% in R-P on the EC dataset. This phenomenon indicates that constraining the variance within a limited range weakens the model's ability to express the degree of uncertainty. Moreover, when either of these two loss functions is ablated, the R@1 metric increases on both the CxC and COCO datasets, with the greatest increase observed when $\mathcal{L}_{\sigma}$ is removed. This phenomenon can be attributed to fundamental differences in annotation quality among the evaluation datasets. Unlike the EC dataset, which benefits from more comprehensive manual annotations, the COCO Caption and CxC datasets contain numerous false negatives. As a result, models may overfit to noisy labels in the two datasets, leading to artificially inflated R@1 scores. This phenomenon aligns with the observations in the ablation results of PCME++ (Chun (2023)), further confirming that Recall@K can yield misleading results on datasets with a substantial number of false negatives. For more ablation results and sensitivity analyses of the hyperparameters $\alpha$, $\beta$, $\gamma$ and the number of Gaussian components $K$, please refer to Appendix E.

### 4.4 QUALITATIVE ANALYSIS

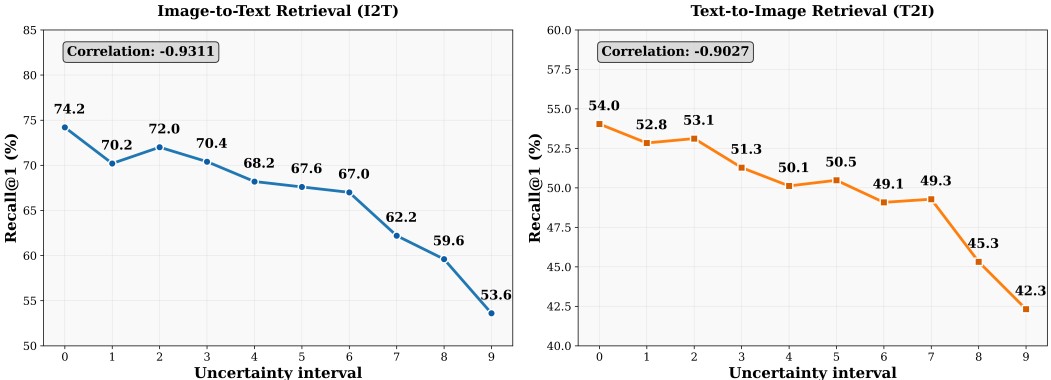

Figure 3: Visualization of relationship between uncertainty and R@1 metric.

**Uncertainty and Accuracy** Figure 3 illustrates the correlation between uncertainty partitioning and retrieval accuracy. For the image-to-text retrieval task, we first compute the uncertainty $\hat{\sigma}(\boldsymbol{s}_{i,j})$ between each image and its most similar text as a confidence measure for that retrieval. Subsequently, all 5000 image retrieval results in COCO test datasets are divided into 10 equal-width intervals based on their confidence scores, and the R@1 accuracy within each interval is calculated. Each interval corresponds to a specific uncertainty range and its associated retrieval accuracy performance. A similar processing approach is applied to the text-to-image retrieval direction. As observed from the figure, both retrieval directions exhibit a significant negative correlation between uncertainty and R@1 (with correlation coefficients of -0.931 and -0.903, respectively). This result indicates that VACSR can assign reasonable uncertainty estimates to retrieval results: when uncertainty is high, the model exhibits lower confidence in its predictions, resulting in correspondingly lower retrieval accuracy; conversely, when uncertainty is low, model confidence is high, and retrieval accuracy improves significantly. Therefore, uncertainty can serve as an effective indicator of the reliability of retrieval results. This observation also underscores VACSR's strong interpretability. For more quantitative analysis results, please refer to Appendix C and G.

## 5 CONCLUSION

This work focuses on addressing the false negative sample problem caused by binary annotations. Unlike previous methods that focus on the inherent ambiguity of data, VACSR directly learning the probabilistic representation of cross-modal similarity to capture fine-grained semantic information by variational inference. Experimental results demonstrate that VACSR effectively improves retrieval accuracy and diversity. Furthermore, VACSR exhibits better performance in out-of-distribution tasks, indicating the broad potential of probabilistic similarity representation in various multimodal downstream tasks.

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

# A   METHOD DETAILS

## A.1   STRAIGHT-THROUGH ESTIMATOR FOR MSE LOSS

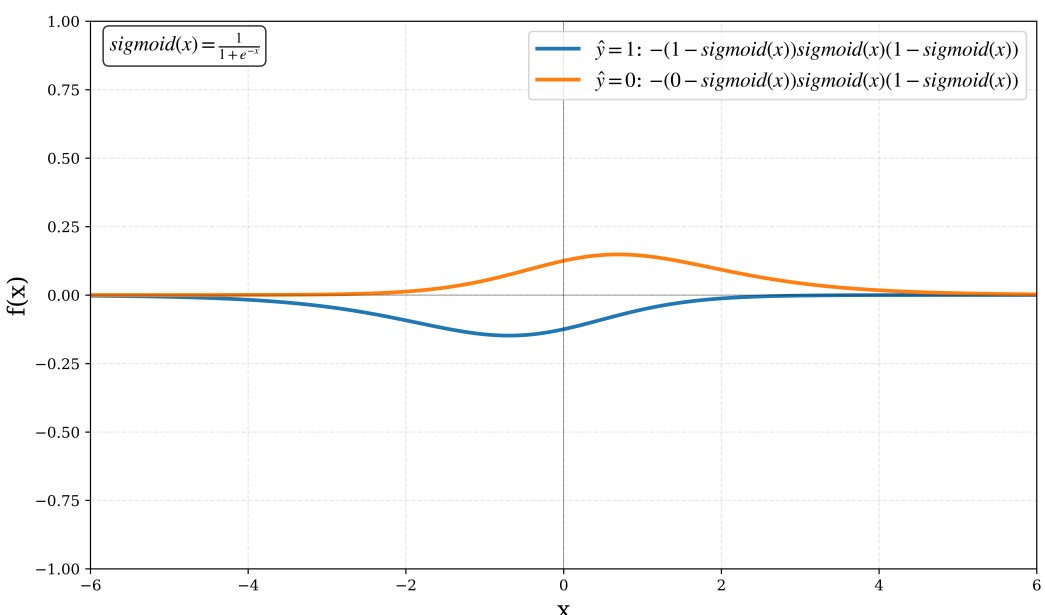

Figure 4: Visualization of MSE loss in classification task.

As described in Section 3.1, we adopt the mean squared error (MSE) loss as the optimization objective. This loss models the generative process as a Gaussian distribution, which simplifies the computation and ensures the continuity and structure of the latent space. However, since the annotations are binary, we need to apply sigmoid normalization to the decoder output $\mu(\mathbf{z}_{i,j})$. In this case, the derivative of Equation 6 with respect to $\mu(\mathbf{z}_{i,j})$ is:

$$\frac{\partial ||\hat{y} - sigmoid(\mu(\mathbf{z}_{i,j}))||^2}{2\partial\mu(\mathbf{z}_{i,j})} = -||\hat{y} - sigmoid(\mu(\mathbf{z}_{i,j}))||sigmoid(\mu(\mathbf{z}_{i,j}))(1 - sigmoid(\mu(\mathbf{z}_{i,j})))$$

(12)

The behavior of this gradient function is illustrated in Figure 4: regardless of the sample's positive or negative nature, the gradient approaches zero when the similarity value is near 0 or 1, thereby hindering effective classification. To address this issue, we employ the Straight-Through Estimator (STE) to approximate the gradient, bypassing the differentiation of the sigmoid function by directly treating the output derivative as the input derivative. Thus, Equation 12 becomes:

$$\frac{\partial ||\hat{y} - sigmoid(\mu(\mathbf{z}_{i,j}))||^2}{2\partial\mu(\mathbf{z}_{i,j})} = -||\hat{y} - sigmoid(\mu(\mathbf{z}_{i,j}))||$$

(13)

this form is equivalent to the gradient of the sigmoid loss.

# B   EXPERIMENTAL DETAILS

## B.1   TASKS AND DATASETS

**Image-Text Retrieval**: Following Chun (2023), we employ COCO Caption (COCO) Chen et al. (2015) along with two extended benchmarks—ECCV Caption (EC) Chun et al. (2022) and CxC Parekh et al. (2020) as the evaluation datasets. EC and CxC are built upon COCO with additional human annotations, significantly mitigating the false negative issue and providing a more reliable assessment of model generalization. Notably, EC corrects the largest number of false negative samples. We report Recall@K (R@K) for all benchmarks. For EC, we additionally report mAP@R and

R-Precision (R-P) to provide a comprehensive evaluation of retrieval diversity and to more reliably reflect the model's true generalization capability.

**Domain Generalization**: Following Zhou et al. (2022a), we trained VACSR on ImageNet Deng et al. (2009) and evaluated on four variant datasets that introduce different domain shifts: ImageNet-V2 Recht et al. (2019), ImageNet-Sketch Wang et al. (2019), ImageNet-A Hendrycks et al. (2021b), and ImageNet-R Hendrycks et al. (2021a). We employ a 16-shot setting and use the template "a photo of a <category>" for the word embeddings. This setup is used to assess the model's generalization and robustness to out-of-distribution data.

**Base-to-Novel Generalization**: This evaluation follows widely adopted protocols Zhou et al. (2022a); Khattak et al. (2023), aiming to assess the model's ability to recognize unseen categories after training on only a subset of classes. We conduct experiments on 11 image classification datasets spanning diverse recognition tasks, including general object recognition (ImageNet Deng et al. (2009), Caltech101 Fei-Fei et al. (2004)), fine-grained recognition (OxfordPets Parkhi et al. (2012), StanfordCars Krause et al. (2013), Flowers102 Nilsback & Zisserman (2008), Food101 Bossard et al. (2014), FGVCAircraft Maji et al. (2013)), scene understanding (SUN397 Xiao et al. (2010)), texture classification (DTD Cimpoi et al. (2014)), satellite image recognition (EuroSAT Helber et al. (2019)), and action classification (UCF101 Soomro et al. (2012)). For each dataset, categories are evenly split into base and novel classes. The model is trained on the base classes using only 16 labeled samples per class in a few-shot setting, and subsequently evaluated on both base and novel test sets. This setup effectively measures the model's adaptation on seen categories as well as its zero-shot generalization ability to unseen novel classes.

## B.2 IMPLEMENTATION DETAILS

The core of our method lies in constructing probabilistic representations of cross-modal similarity rather than feature extraction itself; therefore, we adopt task-specific backbone networks for different tasks. For image-text retrieval, we follow the PCME++ framework Chun (2023), utilizing a pre-trained CLIP as the backbone and employing the Generalized Pooling Operator (GPO) Chen et al. (2021) for feature aggregation. To improve efficiency, we replace the original variance prediction Transformer module with a two-layer MLP-based variational adapter. Experiments are conducted with two visual encoders, ViT-B/32 and ViT-B/16, using the AdamP optimizer for 25 epochs, with an initial learning rate of 0.0005 decayed to 10% at epoch 15.

For out-of-distribution generalization tasks, we follow the experimental setup of Yang et al. (2024), using the full CLIP model as the backbone and omitting structures such as GPO. In domain generalization, to prevent overfitting caused by the over-parameterization of CLIP and limited training samples, we fine-tune only the first two layers of the image encoder and the first three layers of the text encoder, training for 5 epochs with cosine annealing learning rate scheduling. For base-to-novel generalization, we conduct hyperparameter search on the number of frozen Transformer layers and training epochs, and adjust the batch size individually for datasets with extreme class distributions (128 for SUN397 and 5 for EuroSAT) to ensure training stability.

## C SIMILARITY DISTRIBUTION

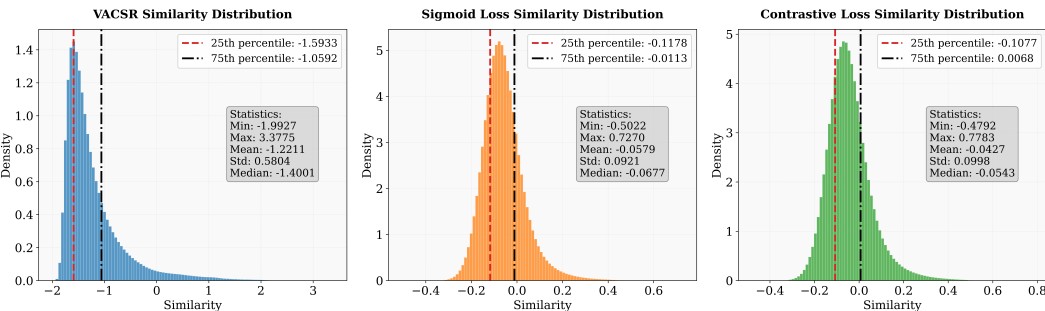

Figure 5: Visualization of similarity.

Figure 5 presents the similarity distributions obtained by VACSR and by fine-tuning CLIP directly using the sigmoid loss ($\mathcal{L}_{sigmoid}$) and the contrastive loss ($\mathcal{L}_{contrast}$). We take the COCO test set as an example, which contains $1.25 \times 10^8$ image-text pairs with a positive-to-negative sample ratio (including false negatives) of $1 : 5000$. Figure 5 shows the similarity distributions derived from $\mathcal{L}_{sigmoid}$ and $\mathcal{L}_{contrast}$ approximate a normal distribution (mean = -0.0579, -0.0427, std = 0.0921, 0.0998), indicating that the similarities of a large number of samples are concentrated in an intermediate range. As discussed in Section 2, false negative samples are subjected to gradients in opposite directions and are thus pushed toward uncertain values rather than receiving confident judgments based on their true semantics. This result makes it difficult to distinguish false negative samples. In contrast, VACSR produces a markedly left-skewed distribution (mean = -1.2211, std = 0.5804). This distribution exhibits a lower central tendency (Median = -1.4001) and greater dispersion, demonstrating that VACSR can more fully span the similarities within the [0,1] interval, thereby effectively modeling the distribution of false negative samples. Furthermore, given the extremely small number of positive samples, the left-skewed distribution aligns more closely with our expectations regarding sample uncertainty. Specifically, the model can now assign lower similarity scores to the majority of negative samples with higher confidence, while reserving higher similarity values for the small number of positive samples. Consequently, VACSR alleviates the false negative sample problem caused by the binary annotations as a whole and achieves more fine-grained similarity modeling.

## D  TOLERANCE OF LOSS FUNCTIONS

Table 7: The impact of false-negative samples on the two types of losses. Here, $L_{sigmoid}$ and $L_{contrast}$ treat the scaling parameter and temperature coefficient as learnable parameters, initialized as $a = 10, b = -10, \tau = 1$. In contrast, $L^*_{sigmoid}$ and $L^*_{contrast}$ fix these parameters to $a = 1, b = 0, \tau = 1$

| | ECCV Caption | | | CxC | COCO | | |
|---|---|---|---|---|---|---|---|
| Loss | mAP@R | R-P | R@1 | R@1 | 1K R@1 | 5K R@1 | RSUM |
| VACSR | **40.7** | **50.1** | **85.5** | **59.7** | **77.2** | **58.1** | **541.4** |
| $L_{sigmoid}$ | 39.3 | 48.7 | 83.1 | 57.3 | 75.5 | 55.6 | 537.0 |
| $L^*_{sigmoid}$ | 0.2 | 0.3 | 0.1 | 0.3 | 0.2 | 0.1 | 1.2 |
| $L_{contrast}$ | 39.0 | 48.7 | 81.7 | 54.9 | 74.0 | 53.0 | 532.6 |
| $L^*_{contrast}$ | 15.7 | 25.5 | 37.5 | 16.4 | 32.1 | 14.8 | 343.2 |

Table 7 presents a comparative experiment of the sigmoid loss and contrastive loss. We observe that the sigmoid loss outperforms the contrastive loss across all evaluation metrics (particularly in R@K results), indicating that its independence during optimization contributes to improved retrieval accuracy. However, when the scaling parameter and temperature coefficient are fixed, the contrastive loss maintains a certain level of performance, whereas the sigmoid loss fails to train altogether. This result highlights the advantage of contrastive loss in terms of tolerance to binary annotations.

## E  SENSITIVITY ANALYSIS

As shown in Figure 6, we systematically analyzes the impact of key parameters on model accuracy, specifically: (1) the sensitivity of the loss function weighting coefficients $\alpha$, $\beta$ and $\gamma$ in Eq.11; and (2) the effect of the number of components $K$ in the Gaussian Mixture Model (GMM) described in Section3.1. To isolate the effect of each variable, all experiments fix non-target parameters to their baseline values and vary only the parameter of interest. It is worth noting that the magnitude of $\alpha$ is significantly smaller than that of the other parameters (e.g., $\beta$ and $\gamma$); therefore, a logarithmic scale is used for the search space of $\alpha$ to ensure its effect is properly evaluated.

### E.1  THE LOSS FUNCTION WEIGHTING COEFFICIENTS

**Hyperparameter** $\alpha$: Analysis of the regularization weight $\alpha$ shows a clear performance peak around $\alpha$=0.0005. In addition, both excessively small (e.g., $\alpha$=0.00005) and large (e.g., $\alpha$=0.01) values of

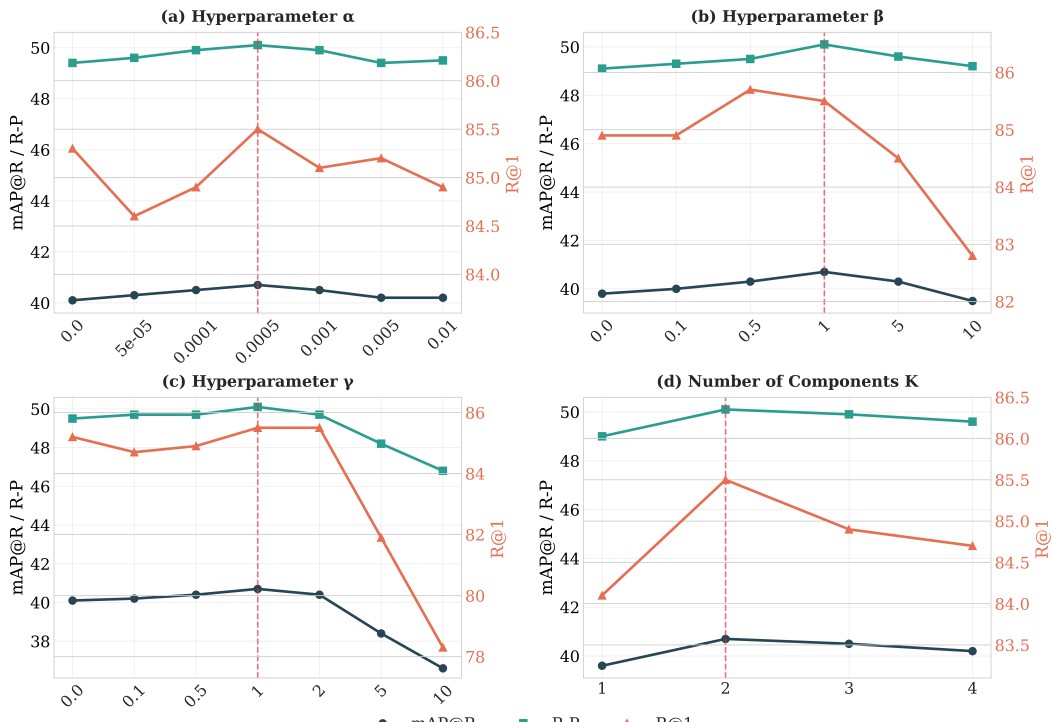

Figure 6: The impact of the loss function weighting coefficients $\alpha$, $\beta$, $\gamma$ and the number of Gaussian components ($K$) on the EC datasets using the CLIP ViT-B/32 backbone. All reported metrics represent the average performance over both image-to-text and text-to-image retrieval directions.

$\alpha$ lead to declines in all metrics. This indicates that appropriate regularization can improve retrieval performance, while overly strong regularization suppresses the model's learning capacity. Notably, when regularization is completely removed ($\alpha$=0.0), the decrease in R@1 is relatively small (85.3), but mAP@R and R-Precision drop significantly to 40.1 and 49.4, respectively. suggesting that regularization constrains the overall quality of retrieval results rather than merely improving Top-1 accuracy.

**Hyperparameter** $\beta$: Setting the weight $\beta$ of the distributional optimization loss to 1.0 yields the best trade-off among all metrics. When $\beta$ is too large (e.g., 10), the excessive emphasis on distributional optimization interferes with the primary learning objective, resulting in a significant drop in R@1 to 82.8. Conversely, when $\beta$ is too small (e.g., 0.1), the variance optimization is insufficient, and mAP@R decreases to 40.0. Additionally, at $\beta$=0.5, R@1 reaches a local optimum of 85.7, but both mAP@R and R-Precision decline, indicating that while the model emphasizes Top-1 accuracy, it may fail to maintain retrieval diversity.

**Hyperparameter** $\gamma$: The model achieves overall optimal performance when the weight $\gamma$ is set to 1.0, while an excessively large $\gamma$ (e.g., 10) leads to catastrophic performance degradation (mAP@R drops to just 36.6). This indicates that over-penalizing negative samples severely harms model performance. In fact, variance optimization for negative samples still uses positive labels, which preserves the model's ability to learn from hard negatives but fails to capture precise uncertainty. This highlights the importance of appropriately weighting negative samples to enhance the model's discriminative power.

In summary, analysis of the three parameters shows that the combination of $\alpha$=0.0005, $\beta$=1.0, and $\gamma$=1.0 achieves the best retrieval performance on the EC dataset (mAP@R 40.7, R-Precision 50.1, R@1 85.5). This configuration ensures sufficient regularization, balanced optimization of positive and negative samples, and avoids the negative effects of excessive penalization.

Table 8: Performance comparison of different methods on base-to-novel generalization across 11 datasets. We employ Clip ViT-B/16 as the encoder backbone. The best results are highlighted in bold.

| Method | Average | | | ImageNet | | | Caltech101 | | | OxfordPets | | |
|---|---|---|---|---|---|---|---|---|---|---|---|---|
| | Base | Novel | HM | Base | Novel | HM | Base | Novel | HM | Base | Novel | HM |
| ZERO-SHOT | 69.34 | 74.22 | 71.70 | 72.43 | 68.14 | 70.22 | 96.84 | 94.00 | 95.40 | 91.17 | 97.26 | 94.12 |
| CoCoOp | 80.47 | 71.69 | 75.83 | 75.98 | 70.43 | 73.10 | 97.96 | 93.81 | 95.84 | 95.20 | 97.69 | 96.43 |
| CLIPOOD | 83.9 | 74.5 | 78.9 | 77.5 | 70.3 | 73.7 | 98.7 | 94.6 | 96.6 | 95.7 | 96.4 | 96.0 |
| MaPLe | 82.28 | 75.14 | 78.55 | 76.66 | 70.54 | 73.47 | 97.74 | 94.36 | 96.02 | 95.43 | 97.76 | 96.58 |
| CoPrompt | 84.00 | **77.23** | 80.48 | 77.67 | 71.27 | 74.33 | 98.27 | 94.90 | 96.55 | 95.67 | **98.10** | **96.87** |
| MMA | 83.20 | 76.80 | 79.87 | 77.31 | 71.00 | 74.02 | 98.40 | 94.00 | 96.15 | 95.40 | 98.07 | 96.72 |
| MMRL | 85.68 | 77.16 | **81.20** | 77.90 | **71.30** | 74.45 | **98.97** | 94.50 | 96.68 | 95.90 | 97.60 | 96.74 |
| VACSR | **85.74** | 76.08 | 80.37 | **78.64** | 70.8 | **74.52** | 98.77 | **95.09** | **96.89** | **96.28** | 97.37 | 96.82 |

| Method | StanfordCars | | | Flowers102 | | | Food101 | | | FGVCAircraft | | |
|---|---|---|---|---|---|---|---|---|---|---|---|---|
| | Base | Novel | HM | Base | Novel | HM | Base | Novel | HM | Base | Novel | HM |
| ZERO-SHOT | 63.37 | 74.89 | 68.65 | 72.08 | 77.80 | 74.83 | 90.10 | 91.22 | 90.66 | 27.19 | 36.29 | 31.09 |
| CoCoOp | 70.49 | 73.59 | 72.01 | 94.87 | 71.75 | 81.71 | 90.70 | 91.29 | 90.99 | 33.41 | 23.71 | 27.74 |
| CLIPOOD | 78.6 | 73.5 | 75.9 | 93.5 | 74.5 | 82.9 | 90.7 | 91.7 | 91.2 | 43.3 | 37.2 | 40.0 |
| MaPLe | 72.94 | 74.00 | 73.47 | 95.92 | 72.46 | 82.56 | 90.71 | 92.05 | 91.38 | 37.44 | 35.61 | 36.50 |
| CoPrompt | 76.97 | 74.40 | 75.66 | 97.27 | 76.60 | 85.71 | **90.73** | **92.07** | **91.4** | 40.20 | 39.33 | 39.76 |
| MMA | 78.50 | 73.10 | 75.70 | 97.77 | 75.93 | 85.48 | 90.13 | 91.30 | 90.71 | 40.57 | 36.33 | 38.33 |
| MMRL | **81.30** | **75.07** | **78.06** | **98.97** | 77.27 | **86.78** | 90.57 | 91.50 | 91.03 | 46.30 | 37.03 | 41.15 |
| VACSR | 79.74 | 73.33 | 76.40 | 98.39 | **77.45** | 86.67 | 90.58 | 91.73 | 91.15 | **47.96** | 36.65 | **41.55** |

| Method | SUN397 | | | DTD | | | EuroSAT | | | UCF101 | | |
|---|---|---|---|---|---|---|---|---|---|---|---|---|
| | Base | Novel | HM | Base | Novel | HM | Base | Novel | HM | Base | Novel | HM |
| ZERO-SHOT | 69.36 | 75.35 | 72.23 | 53.24 | 59.90 | 56.37 | 56.48 | 64.05 | 60.03 | 70.53 | 77.50 | 73.85 |
| CoCoOp | 79.74 | 76.86 | 78.27 | 77.01 | 56.00 | 64.85 | 87.49 | 60.04 | 71.21 | 82.33 | 73.45 | 77.64 |
| CLIPOOD | 81.0 | 79.3 | 80.2 | 80.8 | 58.6 | 67.9 | **97.5** | 64.1 | 77.3 | 85.7 | 79.3 | 82.4 |
| MaPLe | 80.82 | 78.70 | 79.75 | 80.36 | 59.18 | 68.16 | 94.07 | 73.23 | 82.35 | 83.00 | 78.66 | 80.77 |
| CoPrompt | 82.63 | **80.03** | 81.31 | 83.13 | 64.73 | 72.79 | 94.60 | 78.57 | 85.84 | 86.90 | 79.57 | 83.07 |
| MMA | 82.27 | 78.57 | 80.38 | 83.20 | 65.63 | 73.38 | 85.46 | **82.34** | 83.87 | 86.23 | 80.03 | 82.20 |
| MM-RL | 83.20 | 79.30 | 81.20 | **85.67** | **65.00** | **73.82** | 95.60 | 80.17 | **87.21** | 88.10 | 80.07 | 83.89 |
| VACSR | **83.23** | 78.57 | 80.83 | 84.56 | 64.79 | 73.36 | 95.93 | 70.74 | 81.43 | **89.04** | **80.31** | **84.45** |

## E.2 THE NUMBER OF GAUSSIAN COMPONENTS

Sensitivity analysis of the component number $K$ demonstrates that this parameter critically affects the model's ability to capture the intrinsic modal structure of the data. When $K = 2$, the model achieves optimal performance on the EC dataset (mAP@R: 40.7, R-P: 50.1, R@1: 85.5), indicating that two Gaussian components effectively represent the similarity distribution patterns. When $K = 1$, all metrics degrade significantly (e.g., mAP@R: 39.6) due to limited expressive capacity, suggesting that a single Gaussian is insufficient to model the data's complex characteristics. Conversely, when $K \geq 3$, performance declines (with mAP@R at 40.2 when $K = 4$), likely because the increased model complexity introduces redundant parameters, leading to overfitting on noisy training data and reduced generalization. As a result, $K = 2$ is identified as the optimal choice, balancing sufficient expressiveness with the avoidance of over-parameterization risks.

# F    BASE-TO-NOVEL GENERALIZATION

This section provides a detailed analysis of the base-to-novel generalization results across 11 datasets, as shown in Table 8, to further investigate the characteristics of VACSR on various recognition tasks.

**Generic Object Recognition Datasets (ImageNet, Caltech101):** On ImageNet, VACSR achieves the highest base class accuracy (78.64) as well as the best harmonic mean (74.52), indicating strong fundamental performance for large-scale generic category recognition. On Caltech101, which contains a greater number of categories, VACSR ranks first in both novel class accuracy (95.09) and harmonic mean (96.89), demonstrating robust generalization to diverse object categories.

**Fine-grained Recognition Datasets (OxfordPets, StanfordCars, Flowers102, Food101, FGV-CAircraft):** For fine-grained tasks, different methods demonstrate varying strengths. VACSR achieves the highest base class accuracy on OxfordPets (96.28) and leads in novel class accuracy on Flowers102 (77.45). MMRL attains the best performance across base, novel, and harmonic mean metrics on StanfordCars, while CoPrompt slightly outperforms others in harmonic mean on Food101. On the highly challenging FGVCAircraft dataset, VACSR achieves the highest base class accuracy (47.96) and harmonic mean (41.55), indicating its advantage in learning representations for highly specialized categories.

**Scene, Texture, and Satellite Image Recognition (SUN397, DTD, EuroSAT):** On the scene dataset SUN397, VACSR achieves the highest base class accuracy (83.23), while CoPrompt leads in novel class accuracy. For the texture dataset DTD, MMRL attains the best performance across multiple metrics. It is worth noting that on the EuroSAT satellite image dataset, the novel class accuracy of all methods is significantly lower than that of the base class. This is primarily due to the extremely limited textual category descriptions in this dataset (only five), which severely constrains the effectiveness of approaches that rely on cross-modal similarity representations and results in a generalization bottleneck.

**Action Recognition Dataset (UCF101):** On the UCF101 action recognition dataset, VACSR demonstrates a clear advantage, achieving the highest base class accuracy (89.04), novel class accuracy (80.31), and harmonic mean (84.45). This indicates its exceptional transfer capability in modeling dynamic semantic content.

Overall, VACSR shows robust base class recognition across most datasets and leads in comprehensive performance on datasets such as Caltech101, FGVCAircraft, and UCF101. Although its novel class generalization occasionally lags behind methods specifically designed for this purpose, its consistently strong and balanced performance validates the high generalizability of similarity representations learned via variational inference.

# G    VISUALIZATION ANALYSIS OF RETRIEVAL RESULTS

The visualization analysis of retrieval results under a 50% noise ratio clearly demonstrates the robustness advantage of the VACSR method in real-world noisy environments. In the image-to-text retrieval task, VACSR is able to more accurately capture the core semantics of the query image. For example, for Query Image 1, all top-5 results retrievaled by VACSR closely align with the theme "A train is shown on the inside of a station," whereas the results from PCME++ include semantic deviations such as "The two trains." Similarly, in the cases of "surfer" and "giraffes," VACSR's results better maintain consistency with the main subject and scene of the query image, while PCME++ returns unrelated entities such as "birds," "elephants," and "people riding some horses." This indicates that PCME++ is more susceptible to noise in the data and more likely to exhibit semantic drift.

In the text-to-image retrieval task, VACSR demonstrates a more precise grasp of the details in the query descriptions. For the text "A group of people are standing next to an elephant emerging from the water," VACSR successfully retrieves multiple images containing the key element "water," whereas most results from PCME++ overlook this detail. Similarly, for "Two ducks swim together in a pond at sunrise," VACSR accurately matches the number of ducks, while PCME++ makes errors in this aspect. These examples indicate that by learning a distributional representation of similarity,

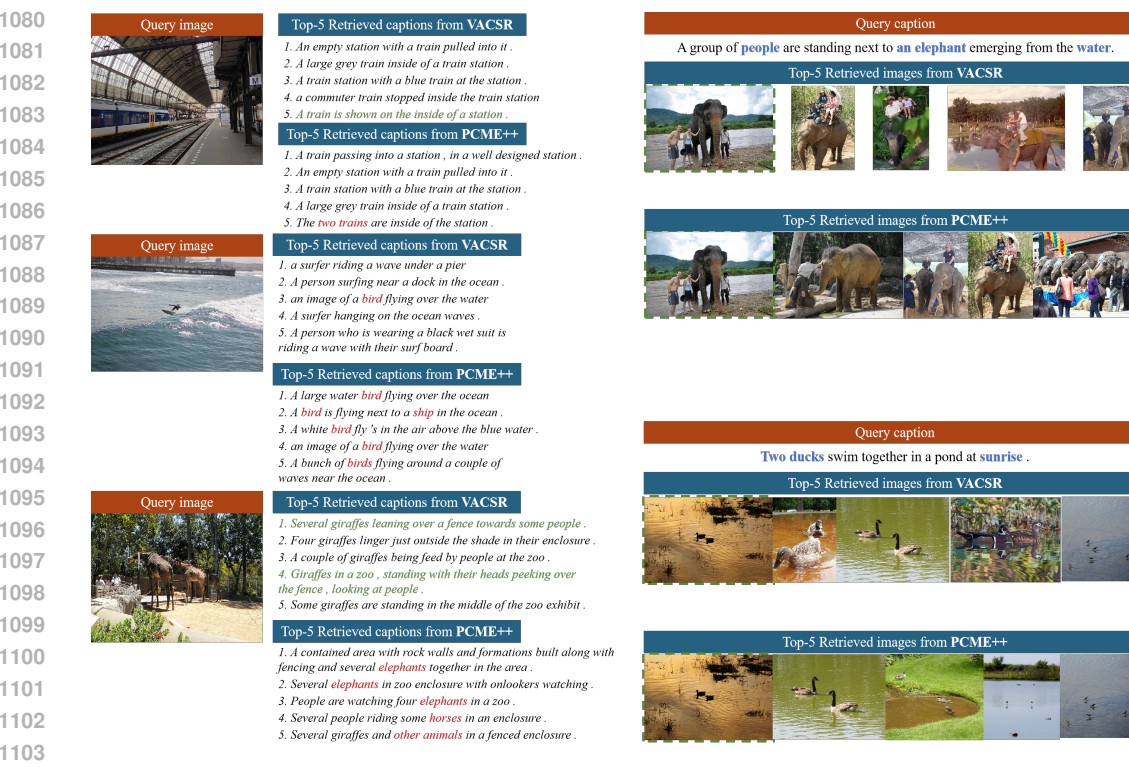

Figure 7: Visualization of retrieval results under a 50% noise ratio. We show the top 5 results retrieved from each image or text query, with the positive samples labeled in the dataset boxed in green.

VACSR is better able to distinguish key semantic features from noisy associations, thereby improving retrieval accuracy.

Notably, in both sets of experiments, many retrieved results were not labeled as "matched" but were highly semantically relevant to the queries. This phenomenon confirms the presence of numerous false negative samples in the original dataset. Both VACSR and PCME++ are able to overcome the limitations of binary sparse annotations and uncover these potential semantic associations.

# H DIFFERENCES BETWEEN VACSR AND PROBABILISTIC EMBEDDING METHODS

Latent variable modelsKingma & Welling (2013) represent sparse one-to-one correspondences as probabilistic mappings in the latent space, providing explicit physical interpretations for different variables. This approach extends the range of semantic expression and captures semantic ambiguity.

In image-text retrieval tasks, the sparsity of binary annotations often fails to accurately capture cross-modal semantic relationships, leading to a large number of false negative samples. To address this, researchers have attempted to learn latent variable distributions by optimizing the probabilistic likelihood of contrastive objectives, in order to model the semantic ambiguity present in the data. The underlying assumption is that, if latent variables follow a certain distribution, data can be mapped to a high-dimensional distributional space via an encoder or decoder, thereby enhancing the expressive power of embedding semantics—resulting in so-called "probabilistic embeddings." For example, Chun et al.Chun et al. (2021) construct a richer embedding space to implicitly capture one-to-many correspondences, where uncertainty estimation can further assist model decision-making. Subsequent workChun (2023) improves the measurement of probability distributions, refines the construction of probabilistic spaces for image-text retrieval, and applies these methods to

uncertainty-based zero-shot classification prompt tuning or pretraining task Ji et al. (2023). Upad-hyay et al.Upadhyay et al. (2023) estimate the embedding distribution of pretrained vision-language models via posterior estimation, avoiding the overhead of retraining large models. Wang et al.Wang et al. (2022) introduce a geometric representation from point to rectangle, retrieving more relevant points within rectangular regions to enhance semantic recall. Li et al.Li et al. (2022) directly opti-mize diversity metrics through differentiable approximation functions, addressing the challenge of non-differentiable objective optimization. Collectively, these approaches expand the set of potential results, constructing a richer semantic retrieval space.

However, existing probabilistic embedding methods typically model images and texts as separate probability distributions. The underlying intuition is that the ambiguity in image-text matching relationships is equivalent to the inherent uncertainty of the image and text data themselves. This implies that even when the data semantic are deterministic, incorrect annotations can still lead to erroneous uncertainty predictions.

The core innovation of VACSR lies in directly representing the matching relationship between image and text data—that is, the cross-modal similarity—as a probability distribution, rather than modeling the distributions of the image or text modalities themselves. On this basis, we propose a distribu-tional optimization loss that replaces the discrete similarity distribution implied by original binary annotations with a continuous, hypothesized probabilistic similarity distribution, thereby avoiding the interference of binary labels. This approach explicitly characterizes the semantic ambiguity in image-text matching relationships, more accurately captures the intrinsic structure of modality associations, and alleviates information loss and misleading constraints caused by sparse binary annotations.

# I    THE USE OF LARGE LANGUAGE MODELS

This paper utilizes DeepSeek Liu et al. (2024) for text polishing and translation assistance.

