# OpenReview forum: "Variational Adapter for Cross-modal Similarity Representation"
_ICLR.cc/2026/Conference — Submitted to ICLR 2026_

### Official Review · Reviewer_vjMJ · 2025-10-20

**Soundness:** 3
**Presentation:** 3
**Contribution:** 2
**Rating:** 6
**Confidence:** 3

**Summary:**

This paper deals with the fact that vision-language models (VLMs) that leaqrn a common space for both texrt and visual inputs are usually trained using sparse, matching/non-matching annotations and discard finer-grained details as well as associations that are missing from the caption (false negatives). The formulate the problem as a form of Variational inference and regularize the space so that overfitting to binary annotations is reduced. They do that via handling uncertainty of the latents: They assign lower uncertainty to positive and  hard negative samples, and lower uncertainty to false negatives. They report strong results on common datasets for cross modal retrieval.

**Strengths:**

The paper deals with the interesting problem of cross-modal retrieval and presents a variant of a probabilistic embeddings for the task that outperform other methods. The method models the uncertainty of false negatives in an intuitive way and shows strong results on common benchmarks, beating related methods. It is interesting that they also show that their probabilistic embeddings help for zero-shot clasisification.

**Weaknesses:**

1. The paper lacks a clear discussion on the relation of this paper to other recent approaches that use probabilistic embeddings. In particular, the relationship between the proposed method and the closely related PCME++ is not sufficiently discussed. The contribution would be significantly strengthened if the authors could clearly articulate the differences—both in design and empirical behavior—in the rebuttal and final version. Moreover, some qualitative results showing cases where the method improves over the top competitor would be interesting to analyse.

2. Building on the previous point, Section 3.1 describes a similarity representation using a 2-component GMM for cross-modal retrieval, following prior work such as Chun et al. (2023) and Bai et al. (2022), as acknowledged by the authors. It remains unclear whether the architecture itself contains any novel elements. Based on the current presentation, the primary novelty appears to lie in the variance loss defined in Eqs. (8)–(10), where the L2 distance to the label is used to modulate the predicted variance. If this is the case, it would be helpful for the authors to clearly emphasize this and clarify what differentiates their approach.

3. The method uses a number of hyperparameters (e.g., $\alpha$, $\beta$, $\gamma$), but their impact on performance is not ablated or discussed. Given the complexity of the setup, some sensitivity analysis or justification for the chosen values would improve the paper's reproducibility and clarity.

**Questions:**

1. Could you clarify the key differences between your approach and PCME/PCME++? The “similarity representation encoder” architecture shown in Figure 2 appears quite similar.
1b. How would PCME++ perform on ImageNet under the same experimental setup as used in Table 3?
1c. What would be the performance of PCME++ if the transformer were replaced with an MLP of the same architecture and size as used in VACSR?

2. Why do you use a single-component GMM? Is there an intuition behind this choice? Was this hyperparameter ablated? What would the performance be with one or three components, keeping everything else the same?

3. How sensitive is the model to the loss hyperparameters $\alpha$, $\beta$, and $\gamma$?

4. Table 4 seems to omit a key ablation on the impact of $L_\sigma$. What would the performance of VACSR if only the $L_\sigma$ component was removed from the loss?

5. On the more challenging and cleaner ECCV Captions dataset, the reported improvements over PCME++ appear to hold primarily for the R@1 metric. Any intuition on why that is?

6. How important is passing the variance though a sigmoid?

---

> ### Author Response · Authors · 2025-12-03
>
> We thank the reviewer for the insightful and constructive comments. We fully agree that systematically clarifying the relationship between our method and existing probabilistic embedding approaches (especially PCME++), conducting thorough hyperparameter analyses, and providing additional empirical results are crucial for the rigor and persuasiveness of the paper.
>
> In response to your concerns, we have made comprehensive additions and improvements in the revised manuscript, Next, we will address your specific questions and suggestions point by point.

---

> ### Author Response · Authors · 2025-12-03
> **Response to Weakness 1**
>
> We would like to thank the reviewers for their valuable comments. In the revised manuscript, **we have systematically addressed the relationship and differences between our method and existing probabilistic embedding approaches (particularly PCME++), as well as supplemented qualitative experimental results, in Appendix H (DIFFERENCES BETWEEN VACSR AND PROBABILISTIC EMBEDDING METHODS) and Appendix G (VISUALIZATION ANALYSIS OF RETRIEVAL RESULTS) of the revised manuscript**, respectively. The specific revisions are as follows.
>
> ***1. Fundamental Differences from Existing Probabilistic Embedding Methods***
>
> Existing probabilistic embedding methods (such as PCME, PCME++, etc.) typically model images and texts themselves as probability distributions, **implicitly assuming that the ambiguity in image-text matching relationships is equivalent to the inherent uncertainty of the image and text data themselves. This implies that even when the data are deterministic, incorrect annotations can still lead to erroneous uncertainty predictions.** In contrast, the core innovation of VACSR is to **probabilistically model the image-text matching relationship—i.e., the similarity—directly, rather than constructing distributions for image or text modalities separately.** Based on this, we **introduce a distributional optimization loss that replaces the discrete similarity distribution implied by binary labels with a continuous hypothesized probability distribution.** This approach more explicitly captures semantic ambiguity in matching relationships, thereby enabling a more precise modeling of the intrinsic structure of cross-modal associations and alleviating information loss caused by annotation sparsity.
>
> ***2. Specific Differences from PCME++ in Design and Empirical Performance***
>
> In terms of methodology, PCME++ continues the approach of learning independent probabilistic embeddings for each modality and enhances modeling capability through improved distribution metrics (such as using the Wasserstein distance). VACSR, on the other hand, introduces a similarity-oriented variational adapter, **whose objective function directly regularizes the similarity distribution rather than adjusting the unimodal embedding distributions.** This design enables VACSR to focus more on learning the uncertainty of matching relationships rather than the uncertainty within the data itself. Empirically, the supplementary visualization analysis (under 50\% noise ratio) clearly demonstrates VACSR’s robustness advantage. In image-to-text retrieval, VACSR’s results more accurately preserve the core semantic consistency with the query image, whereas PCME++ is more susceptible to co-occurrence noise (e.g., mistakenly retrieving numerous “bird”-related texts for a “surfer” query). In text-to-image retrieval, VACSR also exhibits more precise capture of query details (such as “elephants in water” or “two ducks”). These qualitative cases indicate that VACSR has stronger capability in modeling noise and ambiguous matching relationships.

---

> ### Author Response · Authors · 2025-12-03
> **Response to Weakness 2**
>
> Thank you for the reviewer’s question. It should be clarified that the core innovation of VACSR does not lie in the GMM component itself, but rather **in the overall architectural design and the proposed distribution optimization loss function.**
>
> Taking the CSD distance proposed by PCME++ as an example:
> $$
> d(\textbf{Z} _ {v},\textbf{Z} _ {t})=\mathbb{E} _ {\textbf{Z} _ {v},\textbf{Z} _ {t}}||\textbf{Z} _ {v}-\textbf{Z} _ {t}||^{2} _ {2}=||\mu _ {v}-\mu _ {t}||^{2} _ {2}+||\sigma _ {v}+\sigma _ {t}||^{2} _ {2}
> $$
> where $\textbf{Z} _ {v},\textbf{Z} _ {t}$ are the latent variables for image and text data, and $\mu _ {v},\mu _ {t},\sigma _ {v},\sigma _ {t}$ denote the predicted means and variances for image and text data, respectively. As stated in the PCME++: "There are two ways to make $\textbf{Z} _ {v}$ and $\textbf{Z} _ {t}$ closer/further; making $\mu _ {v}$ and $\mu _ {t}$ closer/further, or making $\sigma _ {v}$ and $\sigma _ {t}$ smaller/larger. Hence, if we assume fixed $\mu _ {v}$ and $\mu _ {t}$, we have to decrease $\sigma _ {v}$ and $\sigma _ {t}$ to minimize $d(\textbf{Z} _ {v},\textbf{Z} _ {t})$; if $\textbf{Z} _ {v}$ and $\textbf{Z} _ {t}$ are a certain positive match, then $\sigma _ {v}$ and $\sigma _ {t}$ will be collapsed to zero (i.e., satisfying the condition (b)), and
> $d(\textbf{Z} _ {v},\textbf{Z} _ {t})$ will become Euclidean distance. On the other hand, if the match between $\textbf{Z} _ {v}$ and $\textbf{Z} _ {t}$
> ambiguous (i.e., label can be either positive or negative), then $\sigma _ {v}$ and $\sigma _ {t}$ will not be collapsed to zero,
> for increasing $d(\textbf{Z} _ {v},\textbf{Z} _ {t})$ for the negative match case".
> **However, there is a key issue in this argument regarding the statement: “if the match between $\textbf{Z} _ {v}$ and $\textbf{Z} _ {t}$
> ambiguous (i.e., label can be either positive or negative), then $\sigma _ {v}$ and $\sigma _ {t}$ will not be collapsed to zero”. In fact, ambiguity in image-text matching does not equate to ambiguity in the image or text data themselves. The ambiguity in matching often arises from annotation errors rather than inherent uncertainty in the data.** Therefore, representing images and texts probabilistically on their own cannot fundamentally address the challenges posed by binary labeling.
>
> The innovation of VACSR lies in directly modeling cross-modal similarity probabilistically, and in designing a distributional optimization loss to replace the discrete distribution of binary labels, thereby capturing the uncertainty in matching relationships more explicitly and accurately. We have provided a detailed comparison of this difference in Appendix H of the revised manuscript (please also refer to our response to Weaknesses 1).
>
> Additionally, to more clearly illustrate this architectural design, we have simplified and refined Figure 2 in the revised manuscript according to the reviewer’s suggestions, with an emphasis on the learning process of similarity representation (see our response to Reviewer 3GQh, Weaknesses 2).

---

> ### Author Response · Authors · 2025-12-03
> **Response to Weakness 3**
>
> We appreciate the reviewer’s important suggestion.
> **We systematically evaluate the impact of these three hyperparameters on model performance in Appendix E: SENSITIVITY ANALYSIS of the revised manuscript.** Specifically, we employ a controlled variable approach, fixing the other parameters at their baseline values ($\alpha$=0.0005, $\beta$=1.0, $\gamma$=1.0), and observe how the evaluation metrics on the ECCV Caption dataset (mAP@R, R-Precision, and R@1) vary as each parameter is changed within a specific range.
>
> ***1. Hyperparameter $\alpha$:*** Model performance peaks around $\alpha$=0.0005. When $\alpha$ is too small (e.g., 0.00005) or too large (e.g., 0.01), all metrics decline, indicating that moderate KL regularization helps improve performance, while excessive regularization suppresses model capability. Notably, when $\alpha$=0.0 (i.e., no regularization), the drop in R@1 is relatively small, but mAP@R and R-Precision decrease significantly to 40.1 and 49.4, respectively, suggesting that this regularization term primarily enhances the overall quality of retrieval results rather than merely optimizing a single ranking metric.
>
> ***2. Hyperparameter $\beta$:*** Setting $\beta$=1.0 achieves the best balance among all metrics. When $\beta$ is too large (e.g., 10), the distribution optimization loss excessively interferes with the main task, causing R@1 to decrease significantly to 82.8. Conversely, when $\beta$ is too small (e.g., 0.1), the effect of variance optimization is weakened, and mAP@R drops to 40.0. These results indicate that an appropriate trade-off between the main task and distribution optimization is necessary.
>
> ***3. Hyperparameter $\gamma$:*** The model achieves optimal overall performance when $\gamma$=1.0. If $\gamma$ is too large (e.g., 10), mAP@R drops sharply to 36.6, indicating that excessive penalization of negative samples is detrimental to model performance. Moreover, since the variance optimization for negative samples still relies on the supervision signal from positive samples and is not precise, $\gamma$ should be carefully tuned to balance the strength of uncertainty modeling.
>
> In summary, **the sensitivity analysis confirms that the parameter combination $\alpha$=0.0005, $\beta$=1.0, $\gamma$=1.0 achieves the best retrieval performance on the EC dataset (mAP@R 40.7, R-Precision 50.1, R@1 85.5).** This configuration ensures effective regularization while balancing the optimization of positive and negative samples, thus avoiding performance degradation due to improper weight settings.
>
> We believe that the supplementary sensitivity analysis enhances our understanding of hyperparameter behavior and improves both the reproducibility of our approach and the rigor of the paper. We appreciate the reviewer’s suggestions, which prompted this in-depth investigation.

---

> ### Author Response · Authors · 2025-12-03
> **Response to Question 1**
>
> Thank you for the reviewer’s thoughtful questions. Our responses to the three issues you raised are as follows:
>
> ***1a. Core Differences from PCME/PCME++.***
> We have provided a detailed explanation of the essential methodological differences between VACSR and PCME++ in Appendix H of the revised manuscript, as well as in our responses to Weaknesses 1 and 2. In brief, the innovation of VACSR lies in directly modeling cross-modal similarity probabilistically (rather than unimodal embeddings), and in proposing a distributional optimization loss to more explicitly address the uncertainty in matching relationships.
>
> ***1b. Performance of PCME++ on the ImageNet Domain Generalization Task.***
> We evaluated the performance of PCME++ on the ImageNet domain generalization task under the same experimental settings (see results in the table below). Due to its heavy reliance on the CSD distance metric—which is difficult to effectively optimize in classification tasks—the model fails to converge (with an average accuracy of only 0.1\%). This indicates that the architectural design of PCME++ is not suitable for domain generalization tasks.
> |        | IMAGENET | V2   | S    | A    | R    | AVG. |
> |--------|----------|------|------|------|------|------|
> | VACSR  | 74.3     | 65.7 | 49.7 | 52.4 | 78.4 | 61.6 |
> | PCME++ | 0.1      | 0.8  | 0.3  | 0.1  | 0.1  | 0.3 |
>
> ***1c. Effect of Replacing the Transformer with an MLP in PCME++.***
> We replaced the Transformer-based variance prediction head in PCME++ with a two-layer MLP, consistent with the architecture used in VACSR. The results are as follows:
>
> With a ViT-B/32 backbone, the MLP-based PCME++ shows a slight performance improvement (e.g., EC R@1 increases from 82.9\% to 84.1\%), indicating that a lightweight MLP can effectively model variance when the parameter budget is limited.
> However, with a ViT-B/16 backbone, the MLP-based version performs slightly worse than the original (86.1\% vs. 86.6\%), suggesting that higher-capacity backbones require more powerful variance modeling.
> In contrast, VACSR consistently outperforms under both settings, demonstrating the greater generalizability of its similarity distribution modeling approach.
>
> | Method          | EC mAP@R      | EC R-P | EC R@1 | CxC R@1 | COCO 1K R@1 | COCO 5K R@1 | COCO RSUM |
> |-----------------|---------------|--------|--------|---------|-------------|-------------|-----------|
> | CLIP ViT-B/32 |
> | VACSR           | 40.7          | 50.1   | 85.5   | 59.7    | 77.2        | 58.1        | 541.4     |
> | PCME++          | 40.2          | 49.8   | 82.9   | 56.8    | 75.5        | 55.2        | 537.3     |
> | PCME++ with MLP | 40.3          | 49.8   | 84.1   | 57.0    | 75.6        | 55.3        | 537.0     |
> | CLIP ViT-B/16 |
> | VACSR           | 42.6          | 51.5   | 88.6   | 64.8    | 80.9        | 63.4        | 551.8     |
> | PCME++          | 42.2          | 51.2   | 86.6   | 62.9    | 79.6        | 61.3        | 548.5     |
> | PCME++ with MLP | 41.8          | 51.1   | 86.1   | 62.7    | 79.7        | 61.2        | 548.9     |

---

> ### Author Response · Authors · 2025-12-03
> **Response to Question 2**
>
> Thank you for raising this important question. In our work, we adopted a two-component Gaussian Mixture Model (GMM) rather than a single-component GMM. This choice was guided by a systematic sensitivity analysis of the number of GMM components $K$. In response to your question, **We have included detailed experimental analyses in Appendix E.2 of the revised manuscript.**
>
> Experiments show that the value of
> $K$ significantly affects model performance. When
> $K$=2, the model achieves the best results on the EC dataset (mAP@R: 40.7, R-P: 50.1, R@1: 85.5), indicating that two Gaussian components effectively capture the potential complex structure in the cross-modal similarity distribution. In contrast, when $K$=1, the model's expressiveness is limited, and all metrics drop notably (e.g., mAP@R decreases to 39.6), suggesting that a single Gaussian is insufficient to fit the data distribution. When
> $K \geq 3$, performance also degrades (e.g., mAP@R drops to 40.2 for $K$=4), likely due to overfitting caused by excessive model complexity in the presence of noisy training data.
>
> Therefore, **$K$=2 is identified as the optimal configuration, striking a good balance between model expressiveness and complexity.** This setting effectively characterizes the similarity distribution while avoiding the risk of over-parameterization.
>
> We thank the reviewers again for their valuable feedback, which has helped us clarify this point.

---

> ### Author Response · Authors · 2025-12-03
> **Response to Question 3**
>
> Thank you to the reviewer for raising the question regarding the sensitivity of loss function hyperparameters. We have provided a detailed explanation of this issue **in our response to Weakness 3.**

---

> ### Author Response · Authors · 2025-12-03
> **Response to Question 4**
>
> Thank you for raising this question. In fact, the impact of the $L _ {\sigma}$ component in the loss function **was systematically analyzed in the ablation study presented in Section 4.3 of the original manuscript.** In addition, we have further supplemented the sensitivity analysis of this component in the revised version; please refer to our response to Weakness 3 for details.

---

> ### Author Response · Authors · 2025-12-03
> **Response to Question 5**
>
> We appreciate the reviewer’s important observation. This phenomenon is closely **related to the fundamental difference between the two methods in handling uncertainty.**
>
> As discussed in Section 1 of the main text, probabilistic embedding methods such as PCME++ alleviate the false negative problem by modeling unimodal uncertainty, but their loss functions still rely on binary sparse labels. As a result, the model may assign high uncertainty to any mislabeled sample pair—even if their semantics are clear. While this strategy can enhance retrieval diversity (as reflected in R-P and mAP@R metrics), it may compromise retrieval accuracy: when many semantically unambiguous pairs are assigned high uncertainty, the ranking of the most relevant results can be adversely affected.
>
> In contrast, VACSR models similarity distributions directly and explicitly distinguishes between "uncertainty in matching relations" and "annotation errors" via its distributional optimization loss. On datasets with high annotation quality such as ECCV Captions, VACSR can more accurately assign low uncertainty to truly semantically related pairs, thus achieving the greatest gains in Top-1 accuracy. At the same time, by avoiding excessive penalization of potentially correct pairs, it also maintains steady improvements in R-P and mAP@R (e.g., +1.2\% and +0.6\% under ViT-B/32).
>
> This phenomenon directly demonstrates the core advantage of VACSR: by modeling similarity distributions explicitly, it enables finer distinction between genuine semantic ambiguity and annotation noise, leading to more accurate retrieval on high-quality datasets.

---

> ### Author Response · Authors · 2025-12-03
> **Response to Question 6**
>
> Thank you for raising this important question. The use of the sigmoid function in the variance prediction is a key design of the VACSR model. This mechanism expands the range of uncertainty expression to better align with the model’s learning objectives. Specifically:
>
> ***1. Expanding the range of uncertainty expression.*** The sigmoid function extends the variance range from [0, 1] to [0, +∞), allowing the model to flexibly represent the full spectrum of confidence, from highly certain to extremely uncertain. If the variance is restricted to a finite interval, the model’s ability to characterize uncertainty is constrained, making it difficult to accommodate the inherent ambiguity in real-world data.
>
> ***2. Impact on the optimization process.*** The free scaling of variance enables the model to dynamically adjust the loss weight based on the predicted confidence. As described in Section 3.2: "Let us consider two extreme cases: when $\hat{\sigma}({s} _ {i,j}) \to 0$, $\mathcal{L} _ {recon}$ optimizes $\hat{\mu}({s} _ {i,j})$ towards binary results. Conversely, when $\hat{\sigma}({s} _ {i,j}) \to \infty$, $\hat{\mu}({s} _ {i,j})$ becomes negligible, and $\mu[\hat{\mu}({s} _ {i,j})+\varepsilon \cdot \hat{\sigma}({s} _ {i,j})]$ turns into Gaussian random noise." **If variance is limited to the finite range of [0, 1], this mechanism for dynamically modulating the optimization strength based on uncertainty will be weakened.**
>
> ***3. Ablation study validation.*** To quantitatively assess the impact of removing the sigmoid function, **we have added corresponding experiments in Section 4.3 (Ablation Study) of the revised manuscript.** Results show that on the EC dataset, removing the variance prediction branch with the sigmoid activation leads to a 1.5\% decrease in mAP@R and a 1.4\% decrease in R-Precision. These findings confirm our analysis: constraining variance within a limited range indeed weakens the model’s ability to express uncertainty, thereby negatively affecting overall performance.
>
> In summary, the introduction of the sigmoid function is a crucial design choice that ensures the model can adequately learn and flexibly regulate uncertainty. We thank the reviewer for this question, which allowed us to clarify the underlying mechanism and provide supporting experimental evidence.

---

### Official Review · Reviewer_EqQH · 2025-11-01

**Soundness:** 2
**Presentation:** 2
**Contribution:** 2
**Rating:** 2
**Confidence:** 4

**Summary:**

This paper introduces VACSR, a method designed to address the limitations of binary annotations in vision-language models. Traditional datasets often label image-text pairs as simply “matched” or “mismatched,” which can generate false negatives and hinder semantic understanding. VACSR reframes image-text matching as a variational inference problem, constructing a latent similarity space that captures fine-grained semantics beyond binary labels. It incorporates a distributional optimization loss to assign uncertainty adaptively—reducing the effect of mislabeled samples while enhancing discrimination for informative negatives. Experiments on COCO, ECCV Caption, and CxC datasets show that VACSR improves both retrieval accuracy and diversity while using fewer parameters than prior probabilistic embedding methods.

**Strengths:**

1. The paper is well-structured. It effectively builds from a theoretical analysis of the limitations of standard losses to a well-explained solution using variational inference.
2. The paper gives rigorous gradient-level analysis of standard losses, which provides a solid theoretical foundation for the proposed method and its novel distributional optimization loss.

**Weaknesses:**

1. The novelty is limited. The core problem addressed by VACSR is learning fine-grained similarities between images and texts, which is fundamentally a local-level image-text matching problem. This area has been extensively studied in the last, representative works such SGRAF [1], IMRAM [2], and PVSE [3], which explicitly model the alignment between image regions and text phrases to capture detailed semantic relationships. Therefore, the paper fails to establish a strong motivation for why a new solution is needed in this research landscape, and the innovation appears incremental rather than foundational.
2. The paper claims to address the false negative issue inherent in binary annotations, which is a form of label noise. However, the experiments lack a controlled noisy correspondence setting, which has been a standard evaluation in prior works like PCME++. Comparisons with other robust learning methods such as NCR [4], Bicro [5], NPC [6] should be added.
3. The construction of the local similarity matrix is fundamentally reliant on the feature representations extracted by the pre-trained CLIP encoder. We all know the pre-trained data of CLIP involves some MSCOCO type data, but how does it work on the out-of-distribution scenarios?

Consequently, the overall performance and generalization of VACSR are heavily reliant on the robustness of CLIP's features. Additional experiments on out-of-domain image-text matching datasets are necessary to demonstrate the generalization capability.

**References:**

[1] Haiwen Diao, et al."Similarity reasoning and filtration for image-text matching," in AAAI, 2021, pp.1218–1226.

[2] Chen Hui, et al. "IMRAM: Iterative matching with recurrent attention memory for cross-modal image-text retrieval." Proceedings of the IEEE/CVF conference on computer vision and pattern recognition. 2020.

[3] Song Yale, et al. "Polysemous visual-semantic embedding for cross-modal retrieval." Proceedings of the IEEE/CVF conference on computer vision and pattern recognition. 2019

[4] Huang et al., “Learning with noisy correspondence for cross-modal matching”, NeurIPS, 2021.

[5] Yang et al., “Bicro: Noisy correspondence rectification for multi-modality data via bi-directional cross-modal similarity consistency”, CVPR, 2023.

[6] Zhang et al., “Negative pre-aware for noisy cross-modal matching”, “AAAI”, 2024.

**Questions:**

Please refer to the weaknesses.

---

> ### Author Response · Authors · 2025-12-03
>
> We sincerely thank the reviewer for their valuable comments. We have carefully considered your insightful suggestions regarding the novelty of the work, the necessity of experiments on noise correspondence, and the effectiveness of pre-trained CLIP features in out-of-distribution scenarios, and have strengthened these aspects in the revised manuscript. Below, we provide detailed point-by-point responses to your specific concerns, along with explanations of the corresponding additions and revisions we have made.

---

> ### Author Response · Authors · 2025-12-03
> **Response to Weakness 1**
>
> We thank the reviewer for their valuable comments. We have carefully considered your suggestions regarding the novelty of our work and would like to provide a clearer explanation below.
>
> ***1. Problem Definition and Core Innovation.***
> You noted that "The core problem addressed by VACSR is learning fine-grained similarities between images and texts, which is fundamentally a local-level image-text matching problem" We would like to clarify this point. **The primary objective of VACSR is not to directly address the local matching problem, but rather to tackle the semantic information loss and false negative issues caused by binary sparse annotations.** As described in Section I, existing datasets (e.g., MS-COCO) adopt a binary “matched/unmatched” annotation paradigm, which forces models to strictly separate all unannotated pairs and overlooks rich semantic relationships among samples, thus impairing model generalization [1][2]. **The key innovation of VACSR is to shift representation learning from modality representations to similarity representations across modalities.** As illustrated in Figure 2, we use variational inference to construct a latent space for cross-modal similarity, **replacing the coarse-grained distribution of ground-truth labels with a hypothesized distribution.** This approach does not rely on local or global feature alignment, but instead regularizes at the similarity distribution level, allowing it to serve as a plug-and-play module for various multimodal tasks.
>
> ***2. Method Generality and Task Extensibility.***
> Unlike methods such as SGRAF, IMRAM, and PVSE, which focus on improving local alignment mechanisms, the strengths of VACSR lie in its task generality and evaluation reliability:
>
> **(1) Task Generality.** Our approach is not limited to image-text retrieval. **We extend it to domain generalization tasks** (assessing model robustness under distribution shifts) and achieve significant improvements, as demonstrated in Section 4.2.
>
> **(2) Evaluation Reliability.** We utilize two extensively annotated benchmarks, ECCV Caption (EC) [1] and CxC [3], along with more robust evaluation metrics such as mAP@R. Notably, Chun et al. (2022) have shown that traditional R@K metrics are unreliable in the presence of annotation noise. These choices allow for a more accurate assessment of the model’s generalization capability.
>
> ***3. Figure Modification and Additional Experiments.***
> To better highlight the core ideas, **we have optimized Figure 2 in the revised manuscript** by removing redundant feature extraction steps and emphasizing the similarity representation learning component. **We have also added extensive experiments to systematically validate the method’s advantages in generalization and robustness to noise:**
>
> **(1) In the noisy association tasks** (Section 4.2 in the revised manuscript), VACSR demonstrates outstanding robustness across different noise levels. For example, with 20\% label noise, VACSR outperforms the second-best method on the ECCV Caption dataset by **6.3\%, 4.2\%, and 4.9\%** in terms of mAP@R, R-P, and R@1, respectively. Even under a high noise ratio of 50\%, the model maintains a leading performance (**It achieves improvements of 10.6\%, 7.2\%, 8.5\%, 20.7\%, 5.3\%, 7.1\%, and 2.1\% over the next-best methods across datasets.**), demonstrating VACSR’s strong tolerance to annotation noise.
>
> **(2) In the base-to-novel generalization evaluation** (Section 4.2 and APPENDIX F in the revised manuscript), VACSR achieves a harmonic mean (H) of 80.37 across 11 datasets, outperforming most baseline methods, and **achieves the best performance in base class recognition (85.74).** These results indicate that the learned similarity representations possess good transferability and do not suffer from performance collapse due to overfitting on base classes.
>
> These experiments collectively demonstrate, from the perspectives of noise robustness and category generalization, that VACSR’s variational inference-based modeling of similarity distributions effectively mitigates the effects of annotation sparsity and noise, enhancing the model’s adaptability in real-world scenarios.
>
> In summary, VACSR fundamentally differs from local alignment methods in problem formulation, methodological design, and task extension. **Its innovation lies in systematically addressing the issue of annotation sparsity at the similarity distribution level, and its generalizability has been demonstrated across a range of generalization tasks.** We appreciate the reviewer’s insightful comments, and we hope that the above clarifications and additional experiments can more clearly demonstrate the contributions of this work.

---

> ### Author Response · Authors · 2025-12-03
> **Response to Weakness 2**
>
> We thank the reviewer for raising this important point. We fully agree on the necessity of noise association experiments and **have added Noisy correspondence experiments in Section 4.2 of the revised manuscript**, providing systematic comparisons with state-of-the-art noise-robust methods such as NCR [4], BiCro[5], and NPC[6].
>
> Experimental results demonstrate that VACSR consistently exhibits strong robustness to noise across different noise levels. Specifically, at a 20\% noise ratio, **VACSR outperforms the second-best method, PCME++, by 6.3\%, 4.2\%, and 4.9\% on the ECCV Caption dataset in terms of mAP@R, R-P, and R@1, respectively; achieves a 12.5\% improvement in R@1 on the CxC dataset; and surpasses NPC on the COCO dataset in 1K R@1, 5K R@1, and RSUM by 4.5\%, 6.1\%, and 1.7\%, respectively. Under a high noise ratio of 50\%, VACSR maintains stable performance, with improvements over the second-best method of 10.6\%, 7.2\%, 8.5\%, 20.7\%, 5.3\%, 7.1\%, and 2.1\% across multiple datasets,** while baseline methods experience significant performance drops under the same conditions, further verifying the model’s strong tolerance to annotation noise.
>
> | Noise ratio | Method | mAP@R | R-P | R@1 | R@1 | 1K R@1 | 5K R@1 | RSUM |
> |:---:|:---|:---:|:---:|:---:|:---:|:---:|:---:|:---:|
> | **20%** | **VSE∞** | 37.0 | 46.3 | 79.7 | 53.6 | 72.0 | 51.8 | 518.6 |
> | | **DAA** | 6.7 | 12.5 | 18.5 | 7.0 | 15.3 | 6.0 | 212.8 |
> | | **PCME** | 37.6 | 47.6 | 79.2 | 50.6 | 70.3 | 48.7 | 520.7 |
> | | **NCR** | 35.9 | 46.0 | 78.0 | 50.6 | 70.1 | 48.8 | 518.6 |
> | | **BiCro** | - | - | - | - | 71.3 | - | 523.2 |
> | | **PCME++** | 37.7 | 47.6 | 80.0 | 52.2 | 71.6 | 50.4 | 524.6 |
> | | **NPC** | - | - | - | - | 73.1 | 53.8 | 529.8 |
> | | **VACSR** | **40.1** | **49.6** | **83.9** | **58.7** | **76.4** | **57.1** | **539.0** |
> | **50%** | **VSE∞** | 18.0 | 28.5 | 43.7 | 20.7 | 39.2 | 19.1 | 394.1 |
> | | **DAA** | 0.3 | 0.8 | 1.0 | 0.3 | 0.8 | 0.2 | 20.9 |
> | | **PCME** | 35.2 | 45.5 | 75.7 | 46.3 | 66.6 | 44.4 | 508.0 |
> | | **NCR** | 34.0 | 44.3 | 75.1 | 47.3 | 66.8 | 45.5 | 508.5 |
> | | **PCME++** | 35.7 | 45.8 | 76.3 | 47.4 | 67.6 | 45.5 | 511.0 |
> | | **NPC** | - | - | - | - | 71.3 | 51.9 | 523.4 |
> | | **VACSR** | **39.5** | **49.1** | **82.8** | **57.2** | **75.1** | **55.6** | **534.2** |
>
> These results indicate that by modeling the latent semantic distribution of cross-modal similarity through variational inference, VACSR effectively **mitigates the impact of noisy labels and demonstrates strong robustness for practical applications.**

---

> ### Author Response · Authors · 2025-12-03
> **Response to Weakness 3**
>
> We thank the reviewer for raising this important question. We understand your concerns regarding the effectiveness of pre-trained CLIP features in out-of-distribution (OOD) scenarios. **It is important to note that our original experimental design already incorporated evaluation of OOD generalization capability,** and to further address your concerns, **we have enhanced this aspect in the revised manuscript.**
>
> ***1. OOD Evaluation Conducted: Domain Generalization.***
>
> **As described in Section 4.2 of the original manuscript, we have evaluated VACSR on the domain generalization task, which focuses on assessing the model’s robustness under distribution shift.** Specifically, the model is trained on ImageNet and tested on four variant datasets (ImageNet-V2, ImageNet-Sketch, ImageNet-A, ImageNet-R), each introducing different types of domain shifts. Experimental results show that VACSR achieves the best performance in all these OOD scenarios (see Table 4), providing initial evidence that the similarity representations built upon CLIP features exhibit strong cross-domain stability.
>
> ***2. Additional OOD Evaluation: Base-to-Novel Generalization.***
>
> **To more comprehensively address your concerns, we have added base-to-novel generalization experiments in the revised manuscript** (see Section 4.2 and Appendix F). This task aims to evaluate the model’s ability to generalize to novel (unseen) classes that were not present during training, which is another important aspect of OOD generalization. Experimental results show that VACSR achieves a harmonic mean (H) of 80.37 across 11 datasets, outperforming most baseline methods (e.g., 75.83 for CoCoOp), and attaining the best performance on base class recognition (85.74). These results indicate that by modeling similarity distributions through variational inference, VACSR can partially overcome reliance on specific class labels, thereby enabling generalization to semantically related unseen classes.
>
> | Method    | BASE  | NEW   | H     |
> |:----------|:-----:|:-----:|:-----:|
> | ZERO-SHOT | 69.34 | 74.22 | 71.70 |
> | CoCoOp   | 80.47 | 71.69 | 75.83 |
> | CLIPOOD  | 83.90 | 74.50 | 78.90 |
> | MaPLe    | 82.28 | 75.14 | 78.55 |
> | CoPrompt | 84.00 | **77.23** | 80.48 |
> | MMA      | 83.20 | 76.80 | 79.87 |
> | MMRL     | 85.68 | 77.16 | **81.20** |
> | VACSR    | **85.74** | 76.08 | 80.37 |
>
> ***3. Clarification on CLIP Pre-training Data and OOD Evaluation.***
>
> You correctly pointed out that the CLIP pre-training dataset includes some MS-COCO-like data. **However, the datasets used in our OOD evaluations—including those for domain generalization (various ImageNet variants) and base-to-novel generalization (11 datasets)—differ significantly from MS-COCO in terms of concepts, domains, and task objectives.** These datasets are specifically chosen to assess model performance under semantic shift and covariate shift. The strong performance of VACSR on these tasks demonstrates that it effectively enhances the model’s generalization ability to scenarios **beyond the scope of the pre-training data distribution.**

---

> ### Author Response · Authors · 2025-12-03
> **Reference**
>
> [1] Chun S, Kim W, Park S, et al. Eccv caption: Correcting false negatives by collecting machine-and-human-verified image-caption associations for ms-coco[C]//European conference on computer vision. Cham: Springer Nature Switzerland, 2022: 1-19.
>
> [2] Gao P, Geng S, Zhang R, et al. Clip-adapter: Better vision-language models with feature adapters[J]. International Journal of Computer Vision, 2024, 132(2): 581-595.
>
> [3] Parekh Z, Baldridge J, Cer D, et al. Crisscrossed captions: Extended intramodal and intermodal semantic similarity judgments for MS-COCO[C]//Proceedings of the 16th Conference of the European Chapter of the Association for Computational Linguistics: Main Volume. 2021: 2855-2870.
>
> [4] Huang Z, Niu G, Liu X, et al. Learning with noisy correspondence for cross-modal matching[J]. Advances in Neural Information Processing Systems, 2021, 34: 29406-29419.
>
> [5] Yang S, Xu Z, Wang K, et al. Bicro: Noisy correspondence rectification for multi-modality data via bi-directional cross-modal similarity consistency[C]//Proceedings of the IEEE/CVF Conference on Computer Vision and Pattern Recognition. 2023: 19883-19892.
>
> [6] Zhang X, Li H, Ye M. Negative pre-aware for noisy cross-modal matching[C]//Proceedings of the AAAI Conference on Artificial Intelligence. 2024, 38(7): 7341-7349.

---

### Official Review · Reviewer_3GQh · 2025-11-11

**Soundness:** 3
**Presentation:** 2
**Contribution:** 2
**Rating:** 4
**Confidence:** 3

**Summary:**

This paper introduces VACSR (Variational Adapter for Cross-Modal Similarity Representation), a method that reformulates image–text matching under sparse binary annotations as a variational inference problem. Instead of treating similarity as a fixed scalar, VACSR models it as a latent Gaussian distribution, thereby capturing fine-grained semantics and mitigating the adverse gradient effects caused by false negative pairs. By optimizing the ELBO over the similarity matrix, the model learns to align the regular dot-product similarity with a distributional representation, which then guides the pair-based sigmoid loss. Evaluation results on three image-text retrieval tasks and out-of-distribution tasks show the effectiveness and generalizability of the proposed method.

**Strengths:**

1. There is a very rich set of experiments that is well executed. It not only thoroughly evaluates the performance of the proposed method, but also investigates multiple aspects of the design.
2. By mapping the point-wise similarity score to a distribution can help capturing non-binary semantics and this paper explores the idea well.
3. The formula proposed in the paper is correct, and the logic is clear and complete.

**Weaknesses:**

1. **Marginal improvements from KL regularization.**
As shown in Table 4, the gain from adding the KL loss is relatively small and in some cases the variant without KL even slightly outperforms the full model. It’ll be better to provide more analysis.
2. **Clarity of Figure 2.**
The model diagram is rather dense, with multiple symbols and blocks in close proximity. It’ll be better to simply it and show the high-level idea more clearly.

**Questions:**

Please refer to weaknesses.

---

> ### Author Response · Authors · 2025-12-03
>
> Thank you very much for your recognition of our work. In addition, we greatly appreciate your valuable suggestions and have made comprehensive revisions to the manuscript accordingly. Specifically, to address the issue of limited gain from KL regularization, **we have added a weight sensitivity analysis as well as an in-depth discussion on noisy labeled data**. Regarding the readability of Figure 2, **we have simplified the diagram structure and improved its layout and annotations**. All changes have been highlighted in blue in the revised manuscript. Below, we provide detailed responses to your specific comments point by point.

---

> ### Author Response · Authors · 2025-12-03
> **Response to Weakness 1**
>
> We thank the reviewer for the valuable comments. We have discussed the phenomenon that “the variant without KL even slightly outperforms the full model” at the end of Section 4.3 in the original manuscript. In further response to your concern regarding the “marginal improvements from KL regularization,” we have provided the following additional analyses and explanations in the revised manuscript:
>
> (1) First, in Appendix E.1, we have included a sensitivity analysis on the regularization weight $\alpha$. The results show that the model achieves optimal performance on the EC dataset when $\alpha$ = 0.0005, while both smaller ($\alpha$ = 0.00005) and larger ($\alpha$ = 0.01) values lead to performance degradation. This demonstrates that an appropriate level of KL regularization is indeed beneficial for improving model performance.
>
> (2) Second, regarding your observation that “the variant without KL even slightly outperforms the full model,” we have provided further analysis in Section 4.3. **We suggest that this phenomenon is closely related to the annotation quality of the datasets.** For instance, the COCO Caption dataset contains a large number of false negative samples (according to Chun et al. [1], approximately 88.2\% of positive images and 72.1\% of positive texts are mislabeled as negatives), and while the CxC dataset is manually cleaned, some noise remains. In contrast, the EC dataset, through stricter manual annotation, significantly mitigates the issue of false negatives. Therefore, on noisier datasets such as CxC and COCO, removing the KL term may result in overfitting to noisy labels and artificially inflated metrics like R@1, which do not reflect actual improvements in generalization.
>
> **(3) Our analysis is also consistent with previous studies.** For example, ablation experiments in PCME++ show that when KL term is removed, **metrics such as CxC R@1 as well as COCO 1K R@1, 5K R@1, and RSUM increase from 56.1, 74.5, 54.3, and 534.5 to 56.7, 75.2, 54.9, and 535.9, respectively.** The authors explicitly state: “Note that COCO and CxC have abundant FNs, hence their R@1 metrics could mislead to a wrong result.” This observation aligns with our findings and further supports our conclusions.

---

> ### Author Response · Authors · 2025-12-03
> **Response to Weakness 2**
>
> Thank you for your valuable suggestions. In response to your comments regarding the readability of Figure 2, we have improved Figure 2 in the revised manuscript and optimized the relevant descriptions in Section 3.1 as follows:
>
> (1) First, we simplified the structure of the figure by removing redundant data input flows, and transferred the dimension annotations (e.g., (B, B, d)) from the figure to Section 3.1, where they are now described in detail in text. At the same time, we have explicitly marked the positions of the regularization term (KL divergence) and the reconstruction term in the figure to help readers intuitively understand the complete ELBO optimization process.
>
> (2) Second, we reorganized the layout of the symbols and modules, increasing the spacing between key components to make the encoder-decoder structure of the variational adaptor clearer. Additionally, we have added detailed explanations for each symbol and module function in both the figure caption and Section 3.1, further improving the figure's readability and comprehensibility.
>
> All modifications have been highlighted in blue in the revised manuscript, aiming to more clearly present the model’s high-level design and the process of variational inference.

---

### Author Response · Authors · 2025-12-03
**Response to the Area Chair: Contributions and Revisions on VACSR**

Dear Area Chair,

This paper proposes a variational inference-based cross-modal similarity representation method, VACSR, which aims to address the semantic information loss and false negative issues caused by binary sparse annotations. Unlike conventional probabilistic embedding methods, VACSR does not model the image or text modality directly; instead, it represents the cross-modal similarity itself as a probability distribution, thereby capturing the uncertainty in matching relationships more explicitly.

During the revision process, we have thoroughly addressed all valuable comments from the reviewers, with a particular focus on the following aspects:

**1. In response to concerns about insufficient discussion of novelty, we have systematically compared the essential differences between VACSR and methods like PCME++ in the Appendix.**

**2. We have added experiments on noise robustness, demonstrating that VACSR significantly outperforms baseline methods under 20\% and 50\% noise ratios (It achieves improvements of 10.6\%, 7.2\%, 8.5\%, 20.7\%, 5.3\%, 7.1\%, and 2.1\% over the next-best methods across datasets under the 50\% noise ratio).**

**3. We have extended the base-to-novel generalization experiments, where VACSR achieves the best performance on base class recognition (85.74).**

**4. We have included additional hyperparameter sensitivity analyses and ablation studies to enhance reproducibility.**

Experimental results show that VACSR **not only achieves excellent performance in image-text retrieval tasks, but also demonstrates strong potential in noise robustness and out-of-distribution generalization.** In particular, our uncertainty analysis reveals a significant negative correlation between retrieval accuracy and predicted uncertainty (with a correlation coefficient around -0.9), highlighting the interpretability of our approach.

Given the high cost and variable quality of multimodal data annotation, VACSR offers a new perspective for building robust cross-modal systems. We believe this work makes a positive contribution to the development of probabilistic representation learning, and respectfully request your reconsideration of the paper’s contribution.

---

### Meta-Review · Area_Chair_aaKV · 2026-01-07

**Summary:**

The primary concern is the unclear contribution. The conceptual and empirical benefit of modelling similarity distributions directly (VACSR) over deriving probabilistic similarity from probabilistic embeddings (PCME++) requires closer examination. There were many missing comparisons that the reviewers introduced in their reviews. While the authors discussed these in their responses, another review cycle would be needed to better judge the advantage of the proposed approach compared to existing ones. The overall assessment does not support acceptance.

**Reviewer Concerns:**

1. Novelty relative to prior probabilistic embedding methods [EqQH, vjMJ]

[Reviewers] All three reviewers questioned whether VACSR's approach of modelling similarity distributions (rather than unimodal uncertainty as in PCME++) represents sufficient novelty.
- EqQH: fundamentally a local-level image-text matching problem (SGRAF, IMRAM, and PVSE).
- vjMJ: architecture appears similar to PCME++.

[Authors] VACSR directly models cross-modal similarity probabilistically rather than constructing distributions for each modality separately. This design explicitly distinguishes between "uncertainty in matching relations" and "annotation errors". Systematic comparison with PCME++ added in Appendix H. VACSR outperforms PCME++ across both ViT-B/32 and ViT-B/16.

[AC] Understands the difference, but incorporating proper comparison (conceptual and empirical) against multiple past technologies will likely require another review cycle.


2. Noise robustness and controlled experiments [EqQH]

[Reviewers] Paper claims to address false negatives (a form of label noise) but lacks controlled noisy correspondence experiments. Compare against robust learning methods like NCR, BiCro, and NPC.

[Authors] Added noisy correspondence experiments in §4.2. Comparisons with NCR, BiCro, and NPC included.

[AC] The noise robustness experiments address the reviewer's concern.


3. Marginal gains from KL regularisation [3GQh, EqQH]

[Reviewers] The gain from KL loss is small (Tab 4). In some cases, the variant without KL slightly outperforms the full model.

[Authors] Provided further supporting experiments. The marginal gains on clean data are attributed to dataset annotation quality: COCO contains 88.2% mislabelled positive images and 72.1% mislabelled positive texts. Removing KL may cause overfitting to noisy labels.

[AC] Makes sense, but the marginal gains on clean data remain a valid concern.


4. OOD generalisation [EqQH]

[Reviewers] Does VACSR work on OOD scenarios, given CLIP's pre-training includes COCO-like data?

[Authors] Domain generalisation experiments already in Section 4.2 (ImageNet-V2, ImageNet-Sketch, ImageNet-A, ImageNet-R). VACSR achieves the best performance across all OOD scenarios.

[AC] The OOD experiments address the concern.


5. Hyperparameter sensitivity and design choices [vjMJ]

[Reviewers] Requested justification for design choices: GMM component selection, loss function hyperparameters, and sigmoid function in variance prediction.

[Authors] Added sensitivity analysis

[AC] Authors provided comprehensive ablation studies to substantiate their design choices. The justifications appear ok.

**Reviewer Scores:**

AC does not find significant factors to change reviewer scores dramatically.

3GQh: 4 > 4
Likely remains unconvinced about marginal KL gains on clean data. The sensitivity analysis confirms the effect is small.

EqQH: 2 > 2 or 4
The core novelty concern is addressed to some degree. The authors have discussed the suggested literature in the responses. However, there are too many of them. Also, it's not so clear why it is a major contribution to model similarity distributions as opposed to deriving the probabilistic similarity based on probabilistic embeddings, an approach taken by prior works.

vjMJ: 6 > 6
No strong reason to increase the score to 8.

---

### Decision · Program_Chairs · 2026-01-26

Reject